# Towards a history of Holocene P dynamics for the northern hemisphere using lake sediment geochemical records

Madeleine Moyle[1], John F. Boyle[1], Richard C. Chiverrell[1]

[1]Department of Geography and Planning, University of Liverpool, Liverpool, L69 7ZT, UK

*Correspondence to*: Madeleine Moyle (M.Moyle@liverpool.ac.uk)

**Abstract**

Present day lake water phosphorus (P) enrichment and accelerated P cycling are changes superimposed on a dynamic Holocene history of landscape development following glaciation, changes in climate, and long-term low-intensity human activity. Knowledge of the history of long-term P dynamics is essential for understanding present-day landscape P export and for managing both terrestrial and aquatic environments. This study is the first attempt to constrain the timing and

magnitude of terrestrial changes in Holocene P dynamics across the Northern Hemisphere using lake sediment records.

Here we reconstruct trajectories in terrestrial Holocene P dynamics for the Northern Hemisphere. We apply a simple process model to published lake sediment geochemical P records from 24 sites, producing records of landscape P yield and reconstructing lake water total phosphorus (TP) concentrations. Individual site trajectories of landscape P yield and lake

water TP vary systematically, with differences attributable to local landscape development history. Three distinct traits are apparent. Mountain sites with minimal direct human impact show falling P supply and conform to conceptual models of natural soil development (Trait 1). Lowland sites where substantial (pre-)historic agriculture was present show progressively increasing P supply (Trait 2). Lowland sites may also show a rapid acceleration in P supply over the last few centuries, where high intensity land use, including settlements and farming, are present (Trait 3). Where data availability permitted

comparison, our reconstructed TP records agree well with monitored lake water TP data, and our sediment inferred P yields are comparable to reported catchment export coefficients. Comparison with diatom inferred TP reveals good agreement for recent records.

Our reconstructions form the first systematic assessment of average terrestrial P export for the Northern Hemisphere over the Holocene and provide the empirical data needed for constraining long-term landscape P cycling models and values for

terrestrial P export that could be used for ocean P cycling models. The long-term perspective provided by our sediment-inferred TP can be used to identify pre-disturbance baselines for lake water quality, information essential to target-driven lake management. We find the first detectable anthropogenic impacts on P cycling ca. 6000 BP, with more substantial impacts as early as 3000 BP. Consequently, to characterise pre-disturbance lake P conditions at Trait 2 and Trait 3 sites it is necessary to consider time periods before the arrival of early farmers. Our use of trait classifications has a predictive power

for sites without sediment records, allowing prediction of TP baselines and P trajectories based on regional landscape development history.

## 1 Introduction

Present day lake water phosphorus (P) enrichment and accelerated P cycling are changes superimposed on a dynamic Holocene history of landscape development following glaciation, changes in climate, and long-term low-intensity human
activity (Boyle et al., 2015; Filippelli and Souch, 1999). Recent moves to mitigate the effects of anthropogenic nutrient enrichment on lakes, through the identification of lake water total phosphorus (TP) reference conditions and targets, requires a thorough quantitative knowledge of this dynamic past, because lake P availability depends ultimately on the terrestrial P cycle. Consequently, a crucial task in palaeolimnology is to provide accurate histories of catchment P yield and lake water TP.

Reconstructed P yield records have application beyond limnology. For example, they can provide the empirical data needed for constraining long-term landscape P cycling models. Furthermore, there are parallels between records of catchment P yield and monitored riverine P flux data, both describing export from terrestrial systems. Values for terrestrial P export are applied to ocean biogeochemical models and are currently supplied by riverine data which lack the long-term (millennial)
temporal perspective of lake sediment P yield records (Sharples et al., 2017).

Total phosphorus records and reference values based on diatom inference models are well established (Battarbee and Bennion, 2012; Hall and Smol, 2010) and extensively applied to quantify TP changes in recent centuries. However, at Holocene time scales diatom TP records are scarce whereas geochemical records of lake sediment phosphorus concentrations
are more numerous. These geochemical records have previously been considered unsuitable for inferring quantitative lake water TP records because lake sediment P concentrations are not correlated with TP concentrations in the overlying lake water (Ginn et al., 2012). However, Moyle and Boyle (2021) show that, at suitable sites, lake-wide average sediment P burial rates provide an unbiased record of long-term average past catchment P yield and lake water TP and their mass balance model is process-realistic, based on simple measurable parameters, and uses a universal calibration.


Here we apply the SI-TP (Sediment inferred TP) model of Moyle and Boyle (2021) to a set of published lake sediment P concentration profiles, generating P yield and sediment inferred lake water TP records. This improves the value of the data for a range of research applications. A total of 24 lake sites were found that had both useable published Holocene P records, and the relevant catchment information needed to apply the mass balance model (Fig. 1). Several other sites with Holocene P
records were excluded for lack of catchment data. While our search will undoubtedly have missed other suitable studies, our

data set has sufficient spatial coverage to start addressing geographical variation in long-term P dynamics, constrained to the Northern Hemisphere due to site locations.

Our objectives were to:

- Compile a data set of lake sediment-derived P yield and lake water TP histories from published sediment geochemical records representing a diverse range of lake sites
- (Moyle and Boyle, 2021) Analyse Holocene P yield and lake water TP trajectories to evaluate natural and anthropogenic controls over long-term variation
- Consider the temporal patterns and likely controls in relation to the quantification of lake water TP reference conditions and restoration targets

## 2 Methods

### 2.1 The SI-TP model

The SI-TP model, described in detail and validated in Moyle and Boyle (2021), is a simple process model based on lake mass balance models developed in the 1970s but reframed from a lake sediment perspective. Lake sediment core data for apparent P burial loading ($L_{core}$, where $L$ refers to a lake-bed normalised flux) is adjusted for focussing effects to estimate the lake-wide mean P sediment burial loading ($L_{sed}$; Eq. (1)). This key variable is then used to estimate the input loading ($L_{in}$; Eq. (2)), and from that the catchment P yield ($P$ $yield$; Eq. (3)), and to estimate past lake water TP ($TP_{lake}$; Eq. (4)) (Moyle and Boyle, 2021). This means that the calculated P yield values come directly from the sediment record and not from a hydrochemical mass balance, i.e. runoff and inflow TP concentration. The phosphorus retention coefficient, $R_P$, is the only model parameter that cannot easily be measured at each site owing to the time and cost of obtaining suitable data. However, it may be predicted using empirical models, and we use three different values to obtain a range of estimated TP and P yield values. The first $R_P$ value is estimated using the model of Kirchner and Dillon (1975). The second and third $R_P$ values are estimated using the Vollenweider (1975) model and are based on two values for apparent settling velocity (v); v = 10, from Vollenweider's original model, and v = 29, representing the Great Lakes, which are known to have exceptionally high apparent settling velocities (Chapra and Dolan, 2012). These represent the typical upper and lower limits of published v values and capture the range of possible TP and P yield reconstructed values. The Moyle and Boyle (2021) SI-TP model has thus a universal calibration and requires no local parameterisation. In the following equations $\overline{z}$ = lake mean depth (m), $z_{core}$ = water depth at coring location (m), $A_L$ = lake area (km$^2$), $A_C$ = terrestrial catchment area (km$^2$), and $q_s$ = the lake water loading (m yr$^{-1}$), equal to the total annual water supply (m$^3$ yr$^{-1}$) divided by the lake area (m$^2$).

$$L_{sed} = L_{Core} \times \frac{\overline{z}}{z_{Core}} \tag{1}$$

$$L_{in} = \frac{L_{sed}}{R_P} \tag{2}$$

$$P\ yield = \ L_{in} \times \frac{A_C}{A_L} \tag{3}$$


$$TP_{lake} = \frac{L_{sed}}{R_P q_s}(1 - R_P) \tag{4}$$

## 2.2 Sediment inferred lake water TP

For the SI-TP model, all outputs represent long-term average values, where long-term means >>annual (i.e. at least multi-annual to decadal scales) but is not precisely defined (Moyle and Boyle, 2021). This constrains the temporal resolution of

any reconstructed record, which therefore cannot represent individual (sub)annual values of lake water TP. The temporal resolution of the reconstructed record is further constrained by the nature of the sediment record on which it is based. $L_{sed}$ represents the long-term average P burial rate, which is subject to smoothing due to both sediment bioturbation and diagenetic migration (Moyle and Boyle, 2021) prior to the signal being locked in (Jilbert et al., 2020). Resolution can vary considerably but we assume typical values in the order of multi-annual to decadal, however in very slowly accumulating

sediments this may be longer. The temporally smoothed nature of the sediment inferred TP record is in sharp contrast with the instantaneous spot values obtained by monitoring, the former representing an attractor value about which the latter varies. In the original model paper, Moyle and Boyle (2021) refer to the sediment inferred long-term average lake water TP as $TP_{lake}$, labelled after the parameter it estimates. However, to avoid confusion with high-resolution monitored TP values (also $TP_{lake}$) we introduce the term SI-TP to represent the long-term average sediment inferred TP values. Analogous to

diatom inferred TP (DI-TP) and chironomid inferred TP (CI-TP) (Brooks et al., 2001), this term aids clarity by distinguishing the smoothed response in the sediment record from the more volatile response of lake water TP. SI-TP can now be substituted for $TP_{lake}$ in Eq. (4).

Throughout this study, SI-TP concentrations are reported in mg m$^{-3}$ rather than µg L$^{-1}$. The two units are numerically identical however the former is preferred in this modelling context because it allows for easier conversion to unit area in

metres.

## 2.3 Data extraction from published records

The lake sediment data set comes from many different publications spanning more than 40 years. These studies vary greatly in formatting, terminology, methodology, and what information is included (including whether data are available as a

supplement or in a repository). In particular, units and naming conventions were found to lack standardization. The

availability of reported geochemical, hydrological, and morphological information needed for this study was also found to vary between the publications, therefore a fully standardised methodology was not possible. Where suitable data were published, standard methods have been used as detailed along with the data references in Tables B1 to B3. However, where available information required an alternative approach, data and calculations are described below on a site-specific basis with further references in Tables B1, B2 and B5.

### 2.3.1 Dry density models

Where water contents have been reported, dry density (g cm$^{-3}$) is given by

$$Dry\ density = (1 - W)/\left(\frac{W}{d_w} + \frac{(1-W)}{d_s}\right) \tag{5}$$

where $W$ = water content (mass fraction), $d_w$ = the density of water (taken as 1 g cm$^{-3}$), and $d_s$ = the particle solid density (typically 2.7 g cm$^{-3}$ for minerogenic sediment).

Several studies did not report either density or water content data, from which dry density can be estimated reliably. We have taken two approaches to estimating dry density data in these cases. First, for sites that reported loss on ignition (LOI) we applied an empirical model based on the association of LOI with dry density (Eq. (6); Fig. A1). Sites in this category include: Lake Windermere, Ennerdale Water, Esthwaite Water, Schulzensee, Dudinghauser See, Tiefer See, and Laguna Zoncho. The relationship between dry density and LOI was established using data from three of the sites reported in this paper (Hatchmere, Immeln, and Ķūži), together with data from Linsley Pond (Livingstone and Boykin, 1962), and Crosemere (Beales, 1980). Dry density (g cm$^{-3}$) was calculated from LOI (wt %) where

$$Dry\ density = 2.13 \times LOI^{-0.682} \tag{6}$$

Second, in the case of Lac d'Anterne, where LOI was not reported, we assumed a density based on an analogue site with carbonate-rich sediment. We assumed the sediment to have 40 % water content based on the typical content at Lac d'Annecy (Loizeau et al., 2001).

### 2.3.2 Lake depth

Where lake mean depth ($\overline{z}$) or core depth ($z_{core}$) were not reported, a default focussing factor ($L_{sed}/L_{core}$) of 0.5 was used to calculate $L_{sed}$, following Boyle et al. (2013b). This rounded value, which assumes the coring took place at the deepest point of the lake, compares well with the study lakes for which we have measured depths (focussing factor 0.47 ± 0.12).

### 2.3.3 Individual sites methodology

Lac d'Annecy

Holocene P data came from Loizeau et al. (2001). The age depth model was created using rbacon (Blaauw and Christen, 2011) with [14]C ages from Brauer and Casanova (2001), with a base depth of 1000 cm. Dry density (g cm$^{-3}$) was calculated from the water content record (Loizeau et al., 2001) using a density of 2.7 g cm$^{-3}$ (calcite) for particulate matter using Eq.

(5). Mass accumulation rate (g m$^{-2}$ yr$^{-1}$) was calculated using reported sediment accumulation rate (cm yr$^{-1}$) from the age depth model output and calculated dry density. Runoff was calculated from $q_s$ using $A_L$ and $A_C$ (Tables B1 and B4).

Lac d'Anterne

Holocene P (P$_2$O$_5$) and sediment accumulation rate data came from Giguet-Covex et al. (2011). To get matching ages, sediment accumulation rate was interpolated in R using approxfun(method = linear) from the 'stats' package (R Core Team, 2020). Dry density was calculated using Eq. (5), assuming the same average sediment water content as Annecy (40 wt %) and a density of 2.7 g cm$^{-3}$ (calcite) for particulate matter. Runoff was calculated from MAP and MAT (Tables B1 and B4) using the equation of Turc (1954).

Tiefer

Holocene P data and sediment accumulation rate came from Selig et al. (2007). LOI also came from Selig et al. (2007) and was interpolated in R using approxfun(method = linear) from the 'stats' package (R Core Team, 2020) to match the depths in the P data. Dry density was calculated using Eq. (6). The age depth model was created using rbacon (Blaauw and Christen, 2011) with $^{14}$C ages from Schwarz (2006). $z_{core}$ was assumed to be equal to $z_{max}$. Runoff was calculated from MAP and MAT (Tables B1 and B4) using the equation of Turc (1954).

Schulzensee

Holocene P data and sediment accumulation rate came from Selig et al. (2007). P concentration was smoothed using a 5-point moving average. LOI also came from Selig et al. (2007) and was interpolated in R using approxfun(method = linear) from the 'stats' package (R Core Team, 2020) to match the depths in the P data. Dry density was calculated using Eq. (6). The age depth model was created in using rbacon (Blaauw and Christen, 2011) with $^{14}$C ages from Schwarz (2006) and the age of Laacher See Tephra (12 900 BP) from Bronk Ramsey et al. (2015). $z_{core}$ was assumed to be equal to $z_{max}$.

Dudinghauser

Holocene P data and sediment accumulation rate came from Selig et al. (2007). LOI also came from Selig et al. (2007) and was interpolated in R using approxfun(method = linear) from the 'stats' package (R Core Team, 2020) to match the depths in the P data. Dry density was calculated using Eq. (6). The age depth model was created using rbacon (Blaauw and Christen, 2011) with $^{14}$C ages from Dreßler et al. (2006). $z_{core}$ was assumed to be equal to $z_{max}$. Runoff was calculated from MAP and MAT (Tables B1 and B4) using the equation of Turc (1954).

Ķūži

Holocene P concentration and mass accumulation rate data came from Terasmaa et al. (2013). $z_{core}$ was assumed to be equal to $z_{max}$. Runoff was calculated from $q_s$ using $A_L$ and $A_C$ (Tables B1 and B4).

### Ennerdale

Holocene P concentration and LOI data came from Mackereth (1966). Dry density was calculated using Eq. (6). The age model assumes constant accumulation and the clay junction at 580 cm was taken to represent the start of the Holocene at 11700 b2k (11650 cal BP). $z_{core}$ was assumed to be equal to $z_{max}$.

### Windermere

Holocene P concentration and LOI data came from Mackereth (1966). Dry density was calculated using Eq. (6). The age model assumes constant accumulation and the clay junction at 450 cm was taken to represent the start of the Holocene at 11700 b2k (11650 BP). $z_{core}$ was assumed to be equal to $z_{max}$.

### Esthwaite

Holocene P concentration and LOI data came from Mackereth (1966). Dry density was calculated using Eq. (6). The age model assumes constant accumulation and the clay junction at 460 cm was taken to represent the start of the Holocene at 11700 b2k (11650 BP). $z_{core}$ was assumed to be equal to $z_{max}$.

### Lake Peipsi

Holocene P concentration data came from Kisand et al. (2017). Water content came from Leeben et al. (2010) and was interpolated in R using approxfun(method = linear) from the 'stats' package (R Core Team, 2020) to match the P depths. This was used to calculate dry density using a density of 2.7 for particulate matter (Eq. (5)). Annual depth equivalent runoff
 (m yr$^{-1}$) was assumed to be the same as the closest river (Stålnacke et al., 1998). The age depth model was created using rbacon (Blaauw and Christen, 2011) with $^{14}$C ages from Leeben et al. (2010) in order to generate depths to calculate sediment accumulation rate.

### Plesne Lake

 We obtained raw data from from Jiri Kopacek (2019, personal communication) for the P concentration data in Kopáček et al. (2007) and mass accumulation rate data in Norton et al. (2016).

### Hatchmere

Holocene P concentration and flux ($L_{core}$) data came from Boyle et al. (2015).

### Sargent Mountain Pond

Holocene P flux ($L_{core}$) data came from Norton et al. (2011). A default focussing factor of 0.5 was used to calculate $L_{sed}$, because $\bar{z}$ is not known. Runoff was calculated from MAP and MAT (Tables B1 and B4) using the equation of Turc (1954).

### Kråkenesvatn

Holocene P flux ($L_{core}$) data are as reported by Boyle et al. (2013b), with additional unpublished sample data from 4000 BP to the recent. A default focussing factor value of 0.5 was used because $\overline{z}$ and $z_{core}$ are poorly known. Runoff (m yr$^{-1}$) was estimated using gridded climate data (averaged between 1971 to 2000) from the Norwegian Meteorological Institute (Norsk Klimaservicesenter, 2021) using the Turc equation (Turc, 1955).

### Immeln

Holocene P concentration, dry density and sediment accumulation rate came from Digerfeldt (1974). The core was assumed to be representative of the entire lake and the full $A_L$ and $A_C$ values were used. A focussing factor of 1 was used to calculate $L_{sed}$ because $z_{core}$ was very similar to the approximate $\overline{z}$. The age depth model was created using rbacon (Blaauw and Christen, 2011) with $^{14}$C ages from Digerfeldt (1974). The age for the top of the core was assumed to be 1970 CE because no coring year was given. Runoff was calculated from MAP and MAT (Tables B1 and B4) using the equation of Turc (1954).

### Trummen

Holocene P flux data came from Digerfeldt (1972). $z_{core}$ was assumed to be equal to $z_{max}$. The output of the Immeln age depth model was used to provide recalibrated ages for the zone boundaries identified in Lake Trummen (Digerfeldt, 1974). Runoff was calculated from MAP and MAT (Tables B1 and B4) using the equation of Turc (1954).

### Jackson Pond

P flux data came from Filippelli and Souch (1999). A default focussing factor value of 0.5 was used because the site is now terrestrialised such that $\overline{z}$ and $z_{core}$ are meaningless here. There is no catchment area data available, so an $A_C : A_L$ ratio of 10 was used for calculation of P yield and SI-TP. Runoff was calculated from MAP and MAT (Tables B1 and B4) using the equation of Turc (1954).

### Anderson Pond

P flux data came from Filippelli and Souch (1999). A default focussing factor value of 0.5 was used because the site is now terrestrialised such that $\overline{z}$ and $z_{core}$ are meaningless here. There is no catchment area data available, so an $A_C : A_L$ ratio of 10 was used for calculation of P yield and lake SI-TP. Runoff was calculated from MAP and MAT (Tables B1 and B4) using the equation of Turc (1954).

### Kokwaskey Lake

P flux data came from Filippelli and Souch (1999). A default focussing factor value of 0.5 was used because $\bar{z}$ and $z_{core}$ are not stated. Runoff was estimated from nearby stream data (Menounos, 2002). $A_C$ is approximated from Souch (2004) and $A_L$ was estimated from map data.

Lower Joffre Lake

Holocene P concentration came from Filippelli et al. (2006) and was interpolated in R using approxfun(method = linear) from the 'stats' package (R Core Team, 2020) to match the dry density intervals. The age depth model was created using rbacon (Blaauw and Christen, 2011) with [14]C ages from Menounos (2002), truncated at 600 cm, and the sediment accumulation rate output from the model was combined with dry density from Menounos (2002) to calculate mass

accumulation rate.

Lake Harris

Holocene P concentration and water content came from Kenney et al. (2016). Water content was used to calculate dry density using a density of 2.7 g cm$^{-3}$ for particulate matter. The age depth model was created using rbacon (Blaauw and

Christen, 2011) with [14]C ages from Kenney et al. (2016) and the sediment accumulation rate output from the model was combined with dry density to calculate mass accumulation rate. An $A_L : A_C$ ratio of 1:10 has been used because $A_C$ is not known. Since $z_{core}$ was not quoted, a value of 7 m was used based on coring location sediment thickness (Kenney et al., 2016) and the likely coring location determined from a map of lake sediment thickness (Danek et al., 1991).

Sämbosjön

Holocene P concentration and flux ($L_{core}$) came from Digerfeldt and Håkansson (1993). A default focussing factor value of 0.5 was used because $\bar{z}$ is not reported. Runoff was calculated from MAP and MAT (Tables B1 and B4) using the equation of Turc (1954).

Dry lake

Holocene P flux ($L_{core}$) data came from Filippelli and Souch (1999). A default focussing factor value of 0.5 was used because $\bar{z}$ and $z_{core}$ are not stated. Runoff (m yr$^{-1}$) was estimated from nearby river flow data (USGS 11051499 Santa Ana R nr Mentone (River Only), CA. (U.S. Geological Survey, 2020)). $A_L$ was estimated from map data.

Laguna Zoncho

Holocene P concentration and inorganic content came from Filippelli et al. (2010). Dry density was calculated from inorganic content (100-LOI) using Eq. (6). The age depth model was created using rbacon (Blaauw and Christen, 2011) with [14]C ages from Clement and Horn (2001) and the sediment accumulation rate output from the model was combined with dry

density to calculate mass accumulation rate. A default focussing factor value of 0.5 was used because $\bar{z}$ is not reported.

Runoff was calculated from MAP and MAT (Tables B1 and B4) using the equation of Turc (1954).

### 2.4 Temporal subdivisions

In addition to analysing the overall patterns in each record, we have separated the records into early, mid, and late Holocene to further analyse change through time. The ages used for these three periods broadly follow the IUGS Holocene subdivisions (Walker et al., 2018). We have further subdivided the late Holocene to separate out the recent period of rapid

change towards the present day, using a boundary at 100 BP (1850 CE), broadly coincident with the end of the Little Ice Age (Grove, 1988). In the case of the recent period, the ages chosen captures the post-1850 increase in TP concentrations seen in many lake records (Battarbee et al., 2011). The four periods used are early Holocene (11,650 BP to 8200 BP), mid Holocene (8200 BP to 4200 BP), late Holocene (4200 BP to 100 BP), and recent period (100 BP to present).

### 3 Results

The 24 lakes in our data set are situated across the Americas and Europe (northwest and central) (Fig. 1) and have a variety of climatic and physical properties (Table B4). The range of lake proprieties is illustrated using a correlation matrix PCA (Fig. 2). The biplot of PC1 and PC2, representing 62 % of the variance, positions the lakes quite well in terms of comparative properties. Thus, low elevation, large lakes with relatively high population density (plotting lower right) contrast with small, high elevation lakes with low direct human impact (upper left). Lakes with warmer climates, lower

runoff, and low water loadings (right) contrast with high water loading, high $A_C : A_L$ lakes (left). The mountain lakes are separated according to climate, with cooler wetter locations to the left, and warm or temperate mountains sites at the top.

The published Holocene P records from the 24 sites vary in their reporting; some sites reported only P flux (n=4), some reported only P concentration (n=16), and some reported both P concentration and P flux records (n=4). The 20 sites with reported Holocene P concentration records are shown in (Fig. 3). The reported or calculated $L_{core}$ records were converted to

lake-wide loading ($L_{sed}$) (Fig. 4). Application of the SI-TP model to the $L_{sed}$ values yields reconstructions of catchment P yield and SI-TP (Fig. 5 and Fig. 6). Comparison of recent period SI-TP with TP monitoring data, available for 16 of the sites (Table B5), yields good agreement (Fig. 7b). Regression on log transformed values gives an adjusted $R^2$ of 0.74 (F = 41.6), with a range of $R^2 = 0.68$ (F = 30.4) to $R^2 = 0.79$ (F= 57.6) based on leave-one-out subsamples, for SI-TP calculated using $R_P$ from the model of Kirchner and Dillon (1975). Consistent with Ginn et al. (2012) there is no relationship between sediment

P concentrations and measured lake water TP (Fig. 7a).

A single site, Dudinghauser See in northern Germany, stands out as having the highest persistent sediment P concentration, with a prolonged spell with values greater than 8 mg g$^{-1}$ (Fig. 3). Dudinghauser also has a cumulative P yield amounting to 2.2 kg P m$^{-2}$ of catchment area (Fig. 8), which is 4 times greater than would typically be present in even glacially reset soil (Boyle et al., 2015), and an order of magnitude greater than would have been supplied from such soil by leaching. It is likely

that our empirical LOI model overestimated the dry density over the high-P interval, illustrating the limitation of not having measured density data. We include the Dudinghauser profile in the figures, but the data are excluded from the statistical analysis.

### 3.1 Classifying P profiles by trait

The P yield profiles (Fig. 5), and particularly the cumulative yield profiles (Fig. 8), reveal considerable variability in shape, demonstrating differing yield histories. Upland sites with no or low historic human impact typically show convex upwards profiles whereas lowland sites with long histories of human activity have concave upwards profiles with mid-Holocene increases, and often a recent acceleration. The high mountain sites that currently have glacial ice in their catchments also show abrupt increases in the mid Holocene. It is possible to identify consistent commonalities and contrasts across most sites in the overall trends in Holocene P yield. These are described here as three traits that are displayed across the sites to a greater or lesser degree.

- Trait 1. Early Holocene P yield maximum
  Sites that show elevated early Holocene P supply appear as convex upwards trends in the cumulative yield plots (Fig. 8). Early P increases are predicted by the Walker and Syers (1976) model of soil evolution, and the effect has been previously described in lake sediments by Filippelli and Souch (1999) and Boyle et al. (2013b), occurring at sites on young landscapes initially rich in geological P. Trait 1 is seen most strongly at Harris, Dry, Sargent Mountain, Ķūži, and Kråkenes, with a weaker profile at Windermere and Ennerdale.

- Trait 2. Mid-Holocene increase in P yield
  Sites that are dominated by a mid-Holocene increase in P supply appear as concave upwards trends in the cumulative yield plots (Fig. 8). This is particularly striking at Dudinghauser, Schulzensee, Tiefer, Hatchmere, and Zoncho, but is also present partially at high mountain sites: Anterne, Kokwaskey and Joffre. The breaks of slope in these curves coincide with independent paleolimnological evidence (e.g. mineral influx, pollen, and charcoal) for landscape disturbance either due to early farming or neoglacial activity, depending on lake location.

- Trait 3. Recent sharp increases in P supply
  Recent (post 1850 CE) increases in P supply are apparent in the cumulative yield plots (Fig. 8) as an abrupt steepening of the cumulative profile in the recent part of the record. Typically, these increases contribute only weakly to the total cumulative P yield record, because while the increase is intense it is also short-lived in Holocene terms. Trait 3 is particularly striking at Trummen, Sämbosjön, Schulzensee, Tiefer, Hatchmere, and Zoncho, and is also present at Harris. These are sites at which farming has continued, intensified by modern practices, or where urban areas have been built.

Anderson and Jackson do not clearly show any of the three traits over the Holocene (Fig. 8). However, on a longer timescale they show convex upwards cumulative P profiles (not shown) after the climate amelioration that follows the Late Glacial Maximum, thus conforming to the Walker and Syers (1976) model (Filippelli and Souch, 1999). They are therefore considered to be Trait 1 sites.

Five lakes do not appear to display any of the traits described above: Annecy, Esthwaite, Immeln, Peipsi, and Plesne (Fig. 8). These sites show neither strong curvature nor clear recent increases in their cumulative P yield profiles. We have not classed these as a separate trait. The reasons for this are explored in the discussion.

Though they differ in relative magnitude due to between-site differences in water loading ($q_s$), the Holocene profiles for SI-TP concentration (Fig. 6) are identical in shape to the P yield profiles (Fig. 5). Consequently, the Holocene dynamics of SI-TP conform to the P yield traits. Trait 1 sites show SI-TP concentration peaking at the start of the Holocene, with a subsequent broadly exponential decline, attributed by Boyle (2007) to apatite weathering and depletion. Trait 2 and 3 sites have low early Holocene SI-TP concentrations, soil apatite having been depleted through prolonged weathering since glacial resetting that long preceded the Holocene (Boyle et al., 2015). Recent human settlements near Trait 3 sites contribute to strongly elevated SI-TP.

## 3.2 Environmental drivers

For each lake site we have estimates for both modern mean annual temperature and modern mean annual depth equivalent runoff (Table B4). Because between-site differences are large relative to likely temporal change through the Holocene, we assume these values are representative of the long-term between-site differences. Comparing site average values for the different time periods, we find a weak negative association of mean P yield with temperature, and weak positive associations of mean P yield with both runoff and altitude in the early and mid Holocene (Fig. 9). These relationships weaken in the late Holocene and recent. In the case of SI-TP concentration (not shown), significant negative associations are seen with temperature, except in the recent period. These relationships are tested further using multiple regression (Tables 1 and 2), which can take combined temperature and runoff effects into account. For all periods, SI-TP is no better fitted by multiple regression than by simple regression. However, P yield gives optimal significant regression models on both runoff and temperature for each interval of the Holocene, with particularly strong associations for the mid Holocene. For runoff, the coefficients are positive, while for temperature the coefficients are negative. For the recent period, no correlation for either is observed. In all time periods, altitude is non-significant in the multiple regression, any effect being captured by runoff and temperature.

## 4 Discussion

### 4.1 Holocene P: concentration vs. flux

Most of the studies used in our analysis report only sediment P concentration records while a smaller number report P flux.
Comparing the P concentration records (Fig. 3) and with those of $L_{sed}$ (P flux normalised to lake area and adjusted for focussing) (Fig. 4), we can see differences in the profile shapes through the Holocene. Concentration is a measure of relative flux, describing an amount per unit mass (e.g. mg g$^{-1}$), and since inputs of both P and sediment to a lake are variable, the changes in P concentration cannot be fully interpreted unless the rate of sedimentation is also known (Anderson et al., 1993; Edmondson, 1974). By turning concentrations into loadings, we get absolute values that tell us about the rate of P supply and
allow us to calculate both past nutrient budgets and past lake water TP concentrations. Following the original application of the SI-TP model by Moyle and Boyle (2021), we also find SI-TP correlates with measured lake water TP (Fig. 7b), giving an R$^2$ of 0.74 on the log transformed data. At the same time, fully consistent with Ginn et al. (2012), we find no relationship between sediment P concentrations and measured lake water TP concentrations (Fig. 7a). Consequently, there is no further discussion of sediment P concentrations.

### 4.2 Holocene P yield magnitude and dynamics

The three traits in catchment P yield we have identified can be interpreted in terms of differing properties of the landscape, including history in relation to glaciation, anthropogenic forcing (particularly farming and settlement), and climate change though the Holocene. Indeed, based on location and an understanding of the landscape history, including human manipulation, the traits apparent at a specific lake can be anticipated. This finding is of potential significance for predicting
regional lake water TP reference conditions that can be used to assess the ecological status of lakes, and could be used alongside current methods (e.g. Battarbee et al., 2011).

The P yield and SI-TP profiles at Trait 1 sites, with a maximum at the start of the Holocene followed by gradual decline through to recent times (Fig. 8), can be attributed to progressive and undisturbed development of rejuvenated soils. For most
400 of these sites the explanation for the rejuvenation is glacial resetting (Boyle, 2007; Engstrom et al., 2000), where glacial or periglacial activity immediately prior to the Holocene has replaced ancient soils by unweathered mineral matter. However, Harris, Jackson, and Anderson are not in such landscapes. In the case of Harris, which is thought to have been a groundwater fed fen before ca. 6000 BP and did not reach its current depth until ca. 2600 BP (Kenney et al., 2016), the SI-TP reconstruction may be regarded as unreliable during the early Holocene. At Anderson and Jackson the early high P flux
cannot be explained by erosional landscape rejuvenation because both sites had boreal forest cover extending back to at least 25 ka BP (Liu et al., 2013). However, both locations received substantial aeolian particle loads during the Last Glacial Maximum (Roberts et al., 2007), and thus experienced depositional rejuvenation. The Walker and Syers (1976) model predicts the Trait 1 convex cumulative trend, as initially high leaching of P from the soil should fall exponentially in the

absence of later disturbance of the soil profile. Thus, sites dominated by Trait 1 occur mainly at high altitude and high latitude, where population densities are low, and farming is minor or absent. At lower altitudes and higher latitudes, disturbance of the soil by human activity, in particular agriculture and ploughing, would have altered the cumulative P profile (Boyle et al., 2015). It is noteworthy that the only lowland mid latitude sites that display Trait 1 (Harris, Anderson, and Jackson) occur in regions that did not experience early farming.

Traits 2 and 3 are found in landscapes that have experienced some disturbance during the Holocene. In the case of Trait 2, this can be either human or climatic disturbance, or a combination of both. At Hatchmere, weak Neolithic and stronger Bronze age increases in P yield are attributed to the substantial impact of early farming, with its dependence on livestock (Boyle et al., 2015). At Dudinghauser, Schulzensee, and Tiefer there is evidence of Neolithic forest clearance, Bronze and Iron Age farming, and all three sites had permanent settlement from the Middle Ages (Selig et al., 2007). At Laguna Zoncho the temporal pattern differs slightly because the step changes in yield occur later in the Holocene than at the European sites. Here the lake is much younger, only forming around 3100 BP, however it too has been impacted by farming, with the entire record considered dominated by human activity (Filippelli et al., 2010). At all these sites it is likely that the mid-Holocene increase in P supply can be explained by human activity disrupting the P cycle.

The three high mountain sites are rather different. At Anterne, in the French Alps, there is a human impact signal; for each of the increases in P yield, Giguet-Covex et al. (2011) also find evidence that points to possible human activity in the catchment, particularly cattle-based farming. However, an increase in minerogenic supply also coincides with the onset of regional Neoglaciation ca. 5500 BP (Giguet-Covex et al., 2011), with a corresponding increase in P yield (Fig. 5), presumably of mineral origin. Similar shifts in supply are also seen coinciding with regional glacial advances between 4600 to 2400 BP and during the Little Ice Age (Giguet-Covex et al., 2011). Thus, increases in P supply in the mid Holocene cannot be attributed purely anthropogenic or climate-driven factors. In contrast, the two British Columbia sites (Kokwaskey and Joffre) certainly experienced no substantial direct human impact. The P increases at various points in the Holocene are attributed to cooler and moister climates leading to increased glacial activity, and the main part of the enhanced P supply is particulate (Filippelli et al., 2006; Filippelli and Souch, 1999). The sediment record at Joffre is dominated by mineral P from glacial sources, particularly during the early Holocene but also the last ca. 3500 years, and the patterns of increasing mineral P fraction seen in the record match known cold events and glacial advances in the area (Filippelli et al., 2006). Kokwaskey also has a high mineral P fraction throughout the record (Filippelli and Souch, 1999). The increases in yield at Kokwaskey (Fig. 5) are attributed to the early Neoglaciation, beginning ca. 7.4 ka in western Canada (Menounos et al., 2009), and increasing mineral P supply from glacial sources. The effects of the Little Ice Age are also identified in the sediment record (Filippelli and Souch, 1999).

Trait 3 sites also occur in landscapes that have experienced disturbance. Here we have separated those lakes that have experienced a recent (post ca.1850 CE), and often very large, acceleration in P yield caused by an increase in human activity. At Harris, yield accelerated as a result of cultural eutrophication after ca. 1900 CE (Kenney et al., 2016), while at Trummen the expansion of the town of Växjö on the lake shore caused extreme nutrient pollution (Digerfeldt, 1972). Were it not for truncated records, Trait 3 would also be present at Esthwaite, Windermere, and Peipsi reflecting 20[th] century contamination by domestic sewage. Sämbosjön shows a particularly strong surface enrichment due partially to farming related nutrient pollution, but also due to surface sediment P cycling (Digerfeldt and Håkansson, 1993) a phenomenon known as a stationary peak where recycled P is stored temporarily near the sediment surface but has no direct bearing on lake P supply history (Moyle and Boyle, 2021). Trait 3 is found at sites suitable for human settlement, whether towns with connected sewers, or rural areas with septic systems. In many cases Traits 2 and 3 are coincident. However, recent changes in upland farming economies, and falling population densities in marginal land, mean that not all Trait 2 affected sites display Trait 3 (Collantes, 2006).

Five lakes do not appear to conform to any of the three traits: Annecy, Esthwaite, Immeln, Peipsi, and Plesne (Fig. 8). These sites are all in relatively low altitude catchments which have not experienced substantial lowland farming and therefore do not show Trait 2 characteristics. At least three of these sites (Annecy, Peipsi, and Esthwaite) might be expected to show Trait 3 if the sediment record had extended to the present day, while the other two sites contain no human settlements and therefore would not show this trait. At all five sites deglaciation occurred some thousands of years before the Holocene which explains why they do not show the Trait 1 characteristic of initially rapid and decreasing P supply over the Holocene. However, these sites are essentially exhibiting the stable phase seen in the mid to late Holocene at the Trait 1 sites and can therefore effectively be considered Trait 1 sites without the initial disturbance. This, along with the absence of any visible human disturbance on terrestrial P cycling (Traits 2 and 3), makes these sites ideal for examining the impacts of natural environmental change on lake–catchment systems, including their use in predicting response to future climate change (Dearing et al., 2006). It is worth noting that the absence of clear Trait 2 signals at these sites need not indicate a lack of anthropogenic disturbance, rather that such signals of human impact on P cycling are dwarfed by natural fluxes to the extent that they are not visible in cumulative P yield profiles (Fig. 8).

### 4.3 Environmental correlations

The three traits capture part of the environmental controls over pathways of P supply to the lakes, particularly the roles for glacially reset soils at high latitudes or altitudes, and for anthropogenic disturbance of the P supply at warmer lowland locations. However, other environmental factors will have influenced the total P supply to each of the lake sites. We anticipate that high values of depth equivalent runoff will promote export of terrestrial P, while high mean annual temperatures associate with and may be a proxy for levels of human activity in temperate landscapes. We do not here consider the role of geology. The success of the morphoedaphic index in predicting lake water TP at sites with low human

impact (Cardoso et al., 2007) suggests that geology plays a role, particularly the presence of limestone, but at present regional scale maps of lithological data are not available, precluding any generalised assessment of bedrock.

The positive association of mean P yield with modern mean annual runoff (Fig. 9, Table 1) for all periods except the recent is consistent with runoff enhanced leaching of soil P playing an important role in P export from terrestrial landscapes (Boyle

et al., 2013b). The breakdown in this relationship in the recent period might be attributed to masking by direct human impacts. For example, the rate of supply of P from domestic sewage is little related to runoff (Withers and Jarvie, 2008). A negative association of P yield with modern mean annual temperature (Fig. 9, Table 1) is also present through the Holocene. Given that a positive association is expected if temperature is a proxy for the levels of local human activity, direct human impact appears not to explain this effect. Instead, this result is consistent with cooler sites being more likely to have

glacially-reset P-enriched soils (e.g. Filippelli and Souch, 1999), and thus more likely to conform to Trait 1. The breakdown in relationship of P yield with both temperature and runoff in the late Holocene (Fig. 9, Table 1) may in part be due to the decreasing availability of apatite after soil development and increasing landscape stability (Boyle et al., 2013b), changing the role of climate to P supply. However, we know that direct human P supply is increasingly important from the mid Holocene and may dominate the total P load. The scale and timing of this impact of human activity will depend on lake location and

land use, with lowland sites much more heavily affected than upland sites (Boyle et al., 2015).

It appears, therefore, that climate associations with P yield are both direct and indirect. Based on river flux data, weathering rate is known to increase with runoff (Amiotte Suchet and Probst, 1993; Bluth and Kump, 1994), with power coefficients of 0.8 for silicates and 1.0 for limestone (Boyle, 2008), so increases in P yield with runoff are expected. Our observed power

coefficients, 0.52 and 0.56 for P yield in the early and mid Holocene respectively, though lower, are close enough to show broad consistency with the observations from contemporary rivers. In contrast, a decrease in weathering rate with temperature is not seen in river studies. Instead, the most likely explanation is that temperature is a proxy for landscape age, low temperature sites being more likely to have experienced glacial resetting and have thus base-rich soils. This means that we have no evidence that a change in temperature would impact the P yield. Rather, of modern anthropogenic drivers, it is

the increasingly intense direct human land use that we would expect to condition future P yield.

Climatic control over SI-TP concentration differs somewhat from that over P yield. The P yield to runoff coefficients (0.52 to 0.56) imply SI-TP to runoff coefficients of -0.44 to -0.48; that is that SI-TP is diluted by increasing runoff. Multiple regression is consistent with this in so far as negative coefficients are obtained. However, the regressions (not shown) are not

statistically significant, so we must conclude that any runoff signal present is weak. The SI-TP data do, however, also show strong negative associations with temperature (Table 2) for the mid and late Holocene which again we must suppose is due to a landscape age association with temperature.

### 4.4 Reference values from the past

Our reconstructions of Holocene P dynamics provide long-term context for present day P supply and lake TP conditions. With these reconstructions we can begin to look at geographic variation in P supply history, and address questions about past lake nutrient status and of natural versus climatic drivers of change. Despite the great site to site variability across the 24 lake records, some generalisation can be made about spatial and temporal variations in past P dynamics by examining mean values associated with the three traits (Fig. 10). To deal with differing sample sizes and skewed datasets, all quoted figures are mean of means on log transformed data. By comparing patterns in the three traits, we can see both the role of climate and the increasing influence of human activity on the P cycle through the Holocene.

### 4.4.1 Holocene landscape P supply

Quantitative reconstruction of past landscape P yield is key to understanding present day terrestrial P cycling, lake water nutrient status, and export of terrestrial P to the oceans. Our identification of Traits associated with specific landscape histories offers the possibility of developing a geographically differentiated method of predicting both present-day lake water TP status and landscape P export at sites where no empirical data exists.

For the lakes in our dataset, sites dominated by Trait 1, those with little or no human impact through most of the Holocene, show progressively decreasing average yield through time (Fig. 10a). These sites have the highest average yield values in the early Holocene of 20 mg m$^{-2}$ yr$^{-1}$, values identical to those reported for modern subglacial soluble reactive phosphorus yield in Greenland (17 to 27 mg m$^{-2}$ yr$^{-1}$ (Hawkings et al., 2016)). In contrast, sites not exhibiting Trait 1 average just 8 mg m$^{-2}$ yr$^{-1}$.

Sites dominated by Trait 2, which are predominantly those with long histories of human impact, show the opposite trend. Progressively increasing average values are seen through the Holocene (Fig. 10a), with a late Holocene average of 30 mg m$^{-2}$ yr$^{-1}$ increasing to 45 mg m$^{-2}$ yr$^{-1}$ by the recent period, compared to those not displaying Trait 2 having averages of 5 mg m$^{-2}$ yr$^{-1}$ and 10 mg m$^{-2}$ yr$^{-1}$, respectively. The late Holocene Trait 2 values are consistent with a long-term P export simulation for typical UK conditions of 30 to 45 mg m$^{-2}$ yr$^{-1}$ (Davies et al., 2016). They are also comparable with phosphorus export coefficients for farmed land-use categories in Scotland, with Improved grassland and Arable land giving values of 32 to 62, and 74 to 154 mg m$^{-2}$ yr$^{-1}$ respectively (Donnelly et al., 2020). The early to mid Holocene Trait 2 values fit well with P yields for average modern low-intensity land use categories, with reported P yields for Montane vegetation of 2.5 to 6 mg m$^{-2}$ yr$^{-1}$ and upland rough grassland and heath of 6 to 12 mg m$^{-2}$ yr$^{-1}$ (Donnelly et al., 2020). These values are much lower than our Trait 1 early Holocene rates, as expected given prolonged history of mineral depletion at such sites (Boyle et al., 2013a).

Finally, the six sites exhibiting Trait 3, those with abrupt increases in recent P yield, have an average recent P yield of 50 mg
$m^{-2}$ $yr^{-1}$, compared with 11 mg $m^{-2}$ $yr^{-1}$ at all other sites. Our Trait 3 rates are low compared with the corresponding export coefficient category (Factories and Urban, 138 to 283 mg $m^{-2}$ $yr^{-1}$; Donnelly et al., 2020), but then none of our catchments are fully occupied by this type of land use. This is not an exhaustive comparison with present day values, for example similar values are reported in an application of the SPARROW model, used to predict TP export and delivery rates, for a catchment in Ontario (Kim et al., 2017), but does show that our P yield records are consistent with modern direct observations.

These reconstructed P yield histories are useful for any attempt to model long-term landscape macronutrient cycling, whether simple process models (Boyle et al 2013b, 2015) or more complex coupled soil-ecosystem models (Davies et al. 2016), and provide an opportunity to critically test our understanding of long-term terrestrial ecosystem dynamics. They also provide spatially distributed estimates of P export to the oceans and, crucially, reconstructions of variability in long-term
(~millennial) trends. Given P availability can impact primary production in the oceans influencing the carbon cycle and global climate (Paytan and McLaughlin, 2007), then better constraint of the terrestrial P yield is of considerable importance (Sharples et al., 2017).

### 4.4.2 Lake water TP concentrations

Reference values play a central role in the management of nutrient impacted lakes, representing the conditions that would be expected in the absence of significant human impact on the lake system. They often form the basis of water quality assessments, for example the EU Water Framework Directive (WFD) (European Commission, 2000), where they can be compared to present day values to test how far a lake has deviated from the ideal "status". Bennion et al. (2011) point out that a distinction must be made between a "pristine" value and a "reference" value as the "pristine" state, the condition of the
lake before any human impact, will vary naturally through time. Lake sediment records provide the opportunity to look at variations in lake condition over long time periods, allowing potential pristine and reference conditions to be identified. Using our Holocene records, we can begin to attempt this here. In separating the lakes by trait, our reconstructed SI-TP values offer an approach for predicting lake nutrient status histories for sites where landscape history is known but no palaeoecological or geochemical data exist for site-specific estimation. By examining whole Holocene trends in lake nutrient
dynamics, we also have an opportunity to identify the earliest human impacts on lake TP concentrations, as well as identifying the periods before any impact where the lakes may be considered pristine. Both inform the debate about levels of lake water TP consistent with "high ecological status", and are crucial for the reliable identification of water quality targets (Hübener et al., 2015).

Trait 1 sites are in catchments with minimal human disturbance, therefore the reconstructed Holocene SI-TP profiles represent naturally shifting pristine conditions. These sites show progressively decreasing mean SI-TP concentrations

through the Holocene (Fig. 10b; 6.4, 4.3, 2.2 mg m$^{-3}$, averages for early, mid, and late Holocene, respectively), whereas those not exhibiting Trait 1 deviate from this pattern (Fig. 10b; 5.8, 4.4, 8.5 mg m$^{-3}$, averages for early, mid, and late Holocene, respectively). The Trait 1 decline is consistent with a dominant role for soil mineral depletion in regulating surface water TP dynamics at undisturbed sites, and the decline in P export predicted by the (Walker and Syers, 1976) conceptual model. These sites also show a much narrower range of mean SI-TP concentrations throughout the Holocene than those sites not exhibiting Trait 1 (Fig. 10b).

Apart from the Trait 2 high mountain sites, the Trait 2 and 3 sites show an increasing impact of human activity on lake nutrient status through the Holocene (Fig. 10b). The eight Trait 2 sites have a recent period mean of 26 mg m$^{-3}$ and a late Holocene mean SI-TP concentration of 17 mg m$^{-3}$, compared to 5 mg m$^{-3}$ in the mid Holocene. Unsurprisingly, the highest mean Holocene SI-TP concentrations are in the recent period at Trait 3 sites; those defined by a sharp increase in modern P supply. These six sites have a recent mean SI-TP concentration of 40 mg m$^{-3}$, compared to 7 mg m$^{-3}$ for the remaining sites (Fig. 10b). This pattern of increasing Holocene P with high modern concentrations is consistent with an observed global increase in lake water TP arising from cultural eutrophication in the last few centuries (Smith, 2003), but also shows there is a much longer history of human impact on lake TP concentrations stretching further back into the Holocene.

The problem of how best to determine the nutrient concentrations expected in a lake with "minimal anthropogenic impairment" is reviewed by Hübener et al. (2015). They compare widely applied empirical modelling approaches with palaeolimnological methods, and particularly consider the issue of how far back in time one must look to find minimal human impact. Battarbee et al. (2011) found the first unambiguous lake sediment evidence of nutrient enrichment linked to human activity from 1850 CE onwards and typically after 1900 CE, an observation supported by Bennion and Simpson, (2011). However, other studies have found evidence of enrichment predating 1850 CE (see Hübener et al., 2015 and Bjerring et al., 2008), particularly in lowland agricultural areas, in some cases suggesting that periods of minimal impact can be found "only by considering century to millennial timescales" (Bradshaw et al., 2006). Considering this, Hübener et al. (2015) cautioned against setting a fixed age for pristine lake conditions, themselves finding the age of initial impairment extending back to ca 1100 CE in four of their fourteen lakes, and beyond the limit of their records (ca 500 CE) in two.

Our Trait 2 lakes, those impacted by early farming, show initial enrichment extending as far back as ca. 6000 BP (Fig. 6), contributing to the evidence for a long history of nutrient enrichment in lowland agricultural areas. The early enrichment is such that reference values based on later periods would be substantial overestimates of minimally impaired conditions. For example, for our Trait 2 sites the late Holocene mean SI-TP of 17 mg m$^{-3}$ is broadly consistent with the mean pre-disturbance DI-TP value (23 mg m$^{-3}$), representing the period from 500 CE to more recent enrichment, found for the northern German sites (Hübener et al., 2015). This suggests that the 1500-year sediment profiles analysis by Hübener et al. (2015), the longest measured DI-TP reference value records to date, still do not extend back to conditions of minimal human

impairment. Of perhaps greater importance is the observation that the model predicted reference values for these lakes (European lake type CB 1; Poikane, 2009) is 24 mg m$^{-3}$, very much greater than our Trait 2 average mid Holocene SI-TP value (5.3 mg m$^{-3}$) and similar instead to our farming-impacted late-Holocene average value (17.2 mg m$^{-3}$). Does this mean that the empirical methods are failing in Trait 2 regions? Proposing a specific CB category predictive model, with higher TP

thresholds than for other European regions, Cardoso et al. (2007) warn that the difference might reflect unaccounted anthropogenic impacts, and recommended further research to resolve the question. Our findings offer tentative evidence that concern about the CB region is well-founded. Two other problematic WFD regions (EC and MED) have also been identified by Poikane et al. (2015). These regions have few lakes in a natural condition and complex climatic and morphological factors that make it difficult to define water quality class boundaries. Poikane et al. (2015) suggest these lakes would also

benefit from the use of palaeolimnological data to define reference conditions, so there is clearly a need for long sediment records in water quality management.

Our findings reinforce the conclusion of Hübener et al. (2015) that defining specific lake states in terms of fixed timescales is unwarranted. However, unlike Hübener et al. (2015), we do not find broad agreement with WDF modelled reference

conditions for the sites in northern Germany, neither do we find agreement for our other Trait 2 sites. While we accept that it is pointless to define lake restoration targets that are unachievable (Bennion et al., 2011), it is nevertheless essential that pristine states are known. We conclude that for lowland lakes in regions subject to prehistoric farming, reference values based on paleolimnological methods require records that extend at least to the mid Holocene in order to capture a period without anthropogenic impairment, and that current WFD modelling methods may be substantially overestimating the "high

status" reference TP values.

### 4.5 Reliability and Limitations

To evaluate the accuracy of our palaeolimnological data we can compare inferred values for recent sediment with corresponding monitored data and compare our results with those of alternative methods. Of the 16 sites where monitored lake water TP data are available, 11 of the SI-TP values lie within a factor of 2 of the measured TP values (Fig. 7b). Where

there is a mismatch between observed and SI-TP, it is typically a function of either 1) a stationary P peak in the surface sediment, 2) low-resolution or missing sediment data, or 3) issues with the accuracy or resolution of the age model constraining the calculated fluxes. From this comparison and previous analysis (Moyle and Boyle, 2021), we conclude that our SI-TP values are unbiased. And from this, given that accurate TP inference depends on the sediment inferred P load and P yield data, we conclude that these too are unbiased.

In principle, a further test of the accuracy of our SI-TP values is provided by comparison with long DI-TP records. This is possible for the whole Holocene at Sargent Mountain Pond using data from Norton et al., 2011, though only at low resolution, and for recent millennia at Schulzensee, Tiefer, and Dudinghauser in Northern Germany using data from Hübener

et al., 2015. At Sargent Mountain Pond, mean Holocene DI-TP is 6.3 mg m$^{-3}$ (Norton et al., 2011), double the value of mean SI-TP (3.5 mg m$^{-3}$). Comparing the profiles of the two records, DI-TP (Norton et al., 2011) does not show the early Holocene peak seen in SI-TP (Figure 6). However, the peak in SI-TP corresponds to a peak in DI-pH (Norton et al., 2011), showing that the diatom assemblage varied in parallel with SI-TP. At the three lowland German lakes the situation is more complex. Averaged across the last 1000 years SI-TP and DI-TP (Hübener et al., 2015) have similar magnitudes at both Tiefer (DI-TP = 22 ± 15, SI-TP = 19 ± 15, mg m$^{-3}$) and Schulzensee (DI-TP = 64 ± 24, SI-TP = 74 ± 28, mg m$^{-3}$). At Schulzensee a large increase in TP concentration is present for both proxies at approximately 1000 BP. The DI-TP record does not extend far enough back in time at Tiefer to test for a comparable increase. At both sites shorter-term variations are less well matched. At Dudinghauser the situation is different. The post 1900 sediments are in reasonable agreement (DI-TP = 38 ± 11, SI-TP = 47 ± 25, mg m$^{-3}$) but for the interval with high sediment P concentration (1600 to 100 BP) there is a poor match (DI-TP = 20 ± 11, SI-TP = 136 ± 63, mg m$^{-3}$). The mismatch at Dudinghauser, and shorter-term differences at Schulzensee and Tiefer may be in part attributed to the use of secondary data where not all parameters required for the SI-TP model (Moyle and Boyle, 2021) are available, in this case sediment dry density data. However, biases in the DI-TP for this older sediment cannot be ruled out. The strong influence of pH on diatom flora may interfere with the DI-TP reconstruction (Juggins et al., 2013), not just at Sargent Mountain Pond, but also at the three German lakes. At these sites, high alkalinity makes the pH particularly sensitive to changes in hypolimnetic carbon dioxide (Stumm and Morgan, 1995), potentially influenced by eutrophication via the impact of algal blooms. Evaluation of these issues of covariance in environmental forcing needs a comprehensive assessment of the diatom ecology, and would benefit from analysis of additional core sites with full physical and geochemical data supported by high-resolution chronologies.

Hatchmere is a good example of a lake sediment record containing a stationary peak; recycled P temporarily stored near the sediment surface. Here, high values for P concentration in the top of the record are unrelated to P supply and are of little use in reconstructing P yield (Moyle and Boyle, 2021). Stationary peaks are an inevitable occurrence at some sites. However, providing they are identified and the peak disregarded, false interpretation can be avoided.

For the purpose of this study, we assume that sediment focusing at each lake is temporally invariant. While it has long been known that sediment focussing intensity varies through a lake's history, and that reductions through the Holocene are likely to have occurred (Davis and Ford, 1982; Likens and Davis, 1975), the possible large variations inferred from modelling (Lehman, 1975) have not be reported. Until a suitable generalised predictive model exists we are compelled, with due caution, to assume constancy (Engstrom and Wright, 1984). We also treat $R_P$ and $q_s$ as constant through the Holocene for two reasons. First, Holocene temporal variation of $R_P$ at our sites is likely to be small compared with between-site differences. Second, the palaeoclimatic information needed to calculate temporally resolved values, such as runoff, is not generally available. The assumption of negligible temporal variability is supported by sensitivity testing: varying $q_s$ (and consequently $R_P$) by 20% results in less than 2% change in $R_P$. However, disregarding temporal variation is not a model

constraint: where larger variations are known, and where reliable palaeoclimate data exist, temporally varying $R_P$ values can be used to drive the SI-TP model (Moyle and Boyle, 2021). Likewise, a correction might be made for the impact of reduced water supply due to abstraction, but this is not thought to have substantially impacted the lakes of this study.

Missing or low-resolution data are an inherent problem with secondary and legacy data sources. Here, the UK Lake District records (Windermere, Ennerdale, and Esthwaite) have not captured the more recent late 20[th] century peak in lake eutrophication. Similarly, the sediment accumulation rate data for the German records (Schulzensee, Dudinghauser See, Tiefer See) were presented as long intervals at a constant rate, masking likely fluctuation or acceleration of accumulation towards the present day. Problems with age models also impact on mass accumulation rate calculations and calculated P yield. At Peipsi, the age model used is based on $^{14}$C ages and the lack of resolution in sediment accumulation rate for the recent period leads to a mismatch between SI-TP and monitored TP values. More problematically, the UK Lake District records (Fig. 1) are dated only by interpolation between basal Holocene and the present, but show how reconstructions can still provide valuable information despite a lack of appropriate dating. For some of the records in this study, a degree of additional post processing of the site data could have achieved better matches to monitored data. For example, we tested developing a better recent sediment accumulation rate record at Peipsi using $^{210}$Pb radiometric dating (Heinsalu et al., 2007) which produces SI-TP values much closer to the monitored TP values. However, our primary aim was to consider the longer record, and emphasis on the more recent would have detracted from this. We also aimed to assess just how much can be learned from even rather incomplete data sets.

Moyle and Boyle (2021) address the question of suitable site choice and data collection, pointing out that this method will not necessarily work everywhere. Nevertheless, in most cases, TP reconstruction using the sediment record can still provide valuable information about long term changes in lake nutrient status and supply. There is a clear need for further data collection at both new and existing study sites, with experimental design built to avoid the issues identified here. For each lake, this should include sufficient appropriately positioned sediment cores with a full suite of measured chemical and physical properties, and robust age models. With proper understanding, potential problems with the sediment record can be anticipated, identified, and avoided.

## 5 Conclusion

Using published sediment P records from 24 lakes distributed across the Northern Hemisphere, we apply the SI-TP model (Moyle and Boyle, 2021), a simple process model, to generate Holocene records of long-term average lake water TP concentration and landscape P yield. Our findings show that long-term P dynamics are controlled by climate, landscape development, and human activity, such that knowledge of individual site development is essential to understanding present day terrestrial P cycling and lake water phosphorus enrichment. We find good agreement between our reconstructed SI-TP

values and contemporaneous monitored lake water TP records. At the three sites with available diatom inferred TP records, we also find comparable results suggesting the geochemical approach can provide an alternative to diatom reconstructions of past water quality and can fill the gap where there is no monitoring data. Our P yield values are comparable with reported export coefficient values for similar landscape types and are thus suitable for long-term landscape P modelling.

Evaluating temporal and spatial variations in the dataset we find distinctive patterns in the terrestrial Holocene P profiles which we use to define three traits, each of which is associated with different catchment properties and land use histories.

- Trait 1 sites show maximal landscape P yields and SI-TP concentrations in the early Holocene and steady subsequent declines. These sites are typically on landscapes reset by glacial activity, at high latitude or altitude, and remote from intense human impact.

- Trait 2 sites have low values for P yield and SI-TP in the early Holocene, and maximal values in the late Holocene and recent. These sites are found in two distinct circumstances. The first is at locations where landscape resetting long preceded the Holocene and where early farming was practiced, usually at lower latitudes and altitudes. The second is at locations at high elevation and without farming, but where climate-induced neoglaciation occurred from the mid Holocene, mobilising P stocks in the catchment.

- Trait 3 sites show strongly increased recent P yields and SI-TP concentrations. These sites are located near modern population centres or in intensive agricultural landscapes.

The long-term SI-TP reconstructions can be used to identify pre-disturbance baselines needed for lake management. Our palaeolimnological approach offers a way of estimating meaningful reference conditions for farming-impacted (Trait 2) areas where current methods are problematic (Cardoso et al., 2007; Poikane et al., 2015). At Trait 2 sites we find initial enrichment as early as ca. 6000 BP, highlighting the need for millennial scale perspective.

This study is a first attempt to constrain the timing and magnitude of changes in terrestrial Holocene P dynamics across the Northern Hemisphere, providing useful long-term average terrestrial P flux and SI-TP values. The sediment record perspective places present day lake P dynamics on a long-term trajectory driven by climate, morphometry, landscape setting, human impact, and time. Organised by traits, this approach has a predictive power for sites without sediment records, allowing estimation of trajectories based on regional landscape development history. By characterising long-term terrestrial exports for the northern hemisphere, the reconstructed values could have application to biogeochemical models that simulate ocean nutrient dynamics on millennial timescales. Improving this data set with further well-dated high-resolution records would enhance the potential contribution of palaeolimnology to both the understanding and management of lake P enrichment, and land export to the oceans.

**Appendices**

Appendix A

Additional figures

Appendix B

Additional tables

**Data availability**

Data output for this study are available at Data Catalogue: https://doi.org/10.17638/datacat.liverpool.ac.uk/1272 (Moyle et al., 2021)

**Author contribution**

The study was conceptualised by all authors. MM led the data collection and processing, model application, and statistical
analysis with input from JB. MM produced the plots and RC produced the map. The first version of the manuscript was written by MM and JB with further improvements and additions by RC.

**Competing interests**

The authors declare that they have no conflict of interest.

**Acknowledgements**

The raw sediment data for Lake Plešné were kindly provided by Jiří Kopáček and Josef Veselý.
The work contained in this paper was conducted as part of PhD research supported by the Natural Environment Research Council (NERC) EAO Doctoral Training Partnership. The studentship was funded by NERC (Grant ref. NE/L002469/1) and Natural England.

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

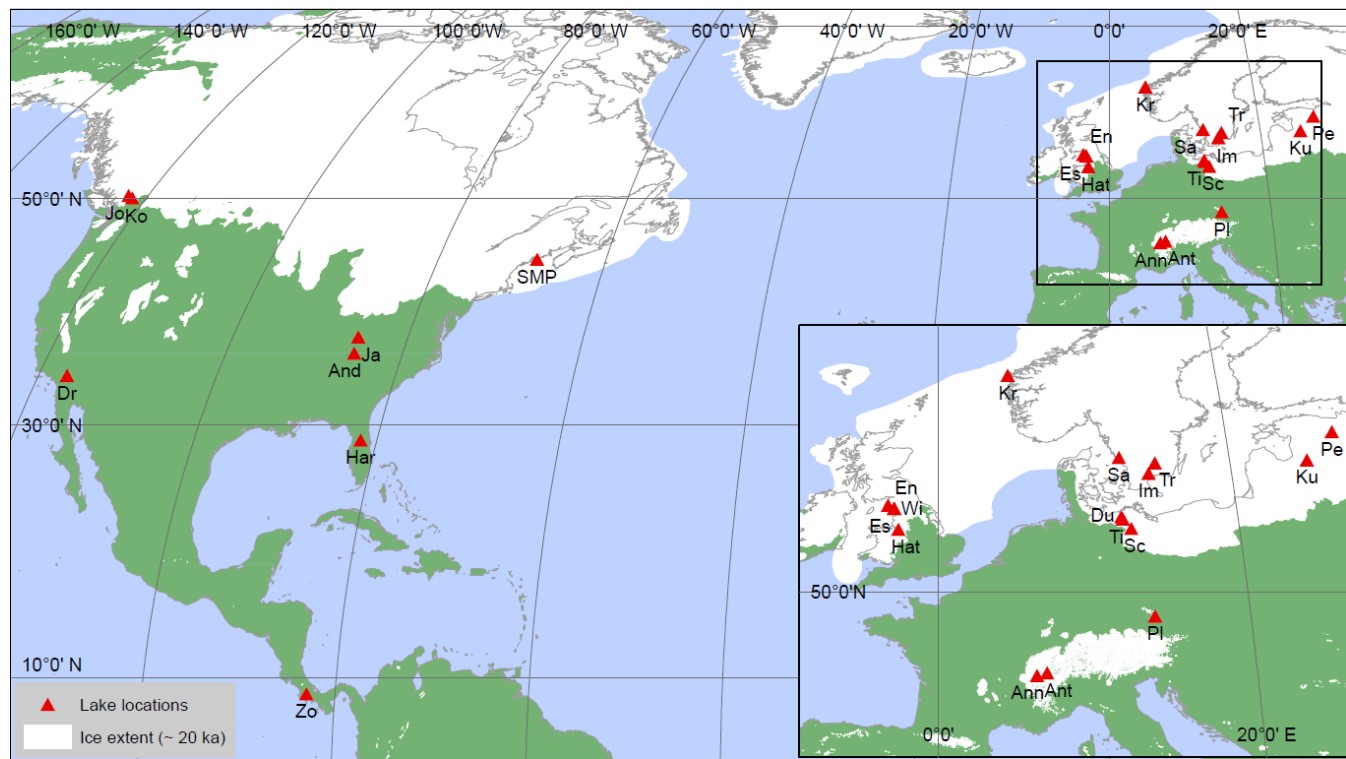

**Figure 1: Spatial distribution of the 24 lake sites: Lac D'Annecy (Ann), Plesne Lake (Pl), Hatchmere, (Hat), Peipsi (Pe), Sargent Mountain Pond (SMP), Dudinghauser (Du), Schulzensee (Sc), Tiefer (Ti), Lac D'Anterne (Ant), Lake Harris (Har), Jackson Pond (Ja), Anderson Pond (And), Dry Lake (Dr), Kokwaskey Lake (Ko), Windermere (Wi), Ennerdale (En), Esthwaite (Es), Kråkenesvatn (Kr), Laguna Zoncho (Zo), Lower Joffre Lake (Jo), Sämbosjön (Sa), Trummen (Tr), Immeln (Im), Ķuži (Ku)**

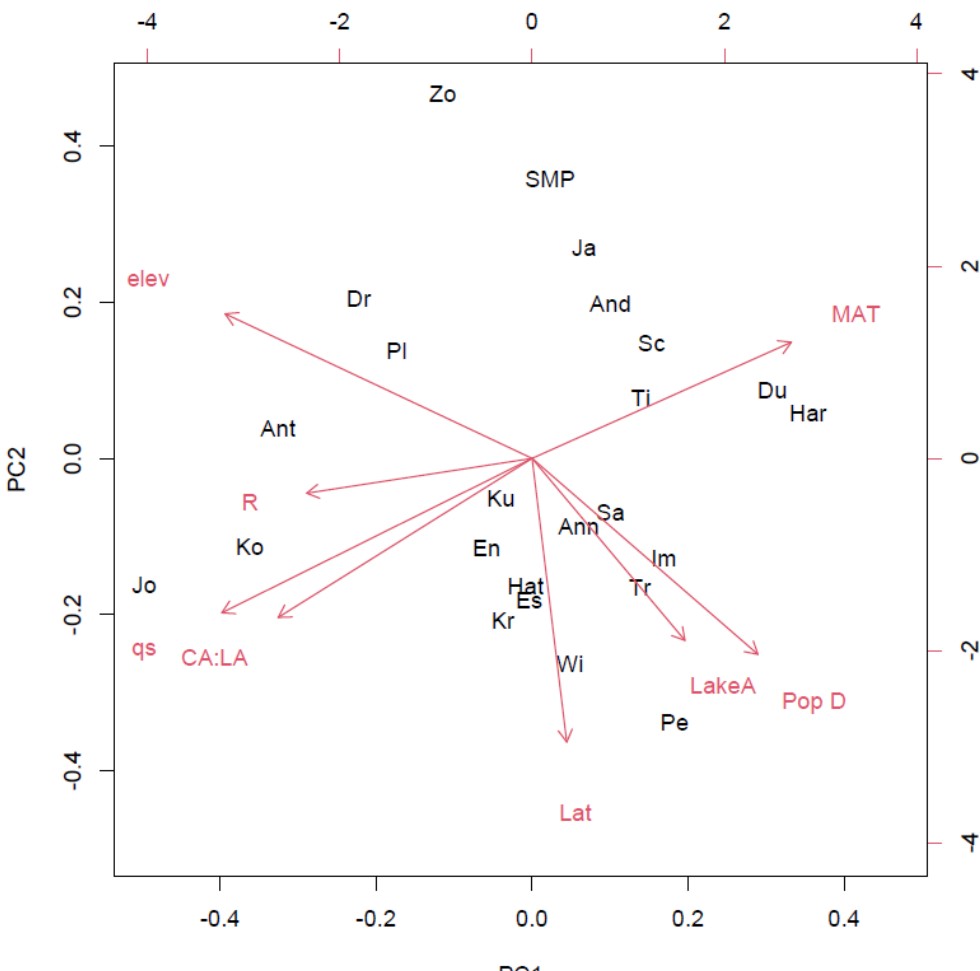

**Figure 2: Correlation matrix PCA biplot illustrating the spatial association of lake properties in the data set. For site abbreviations see Fig. 1 caption**

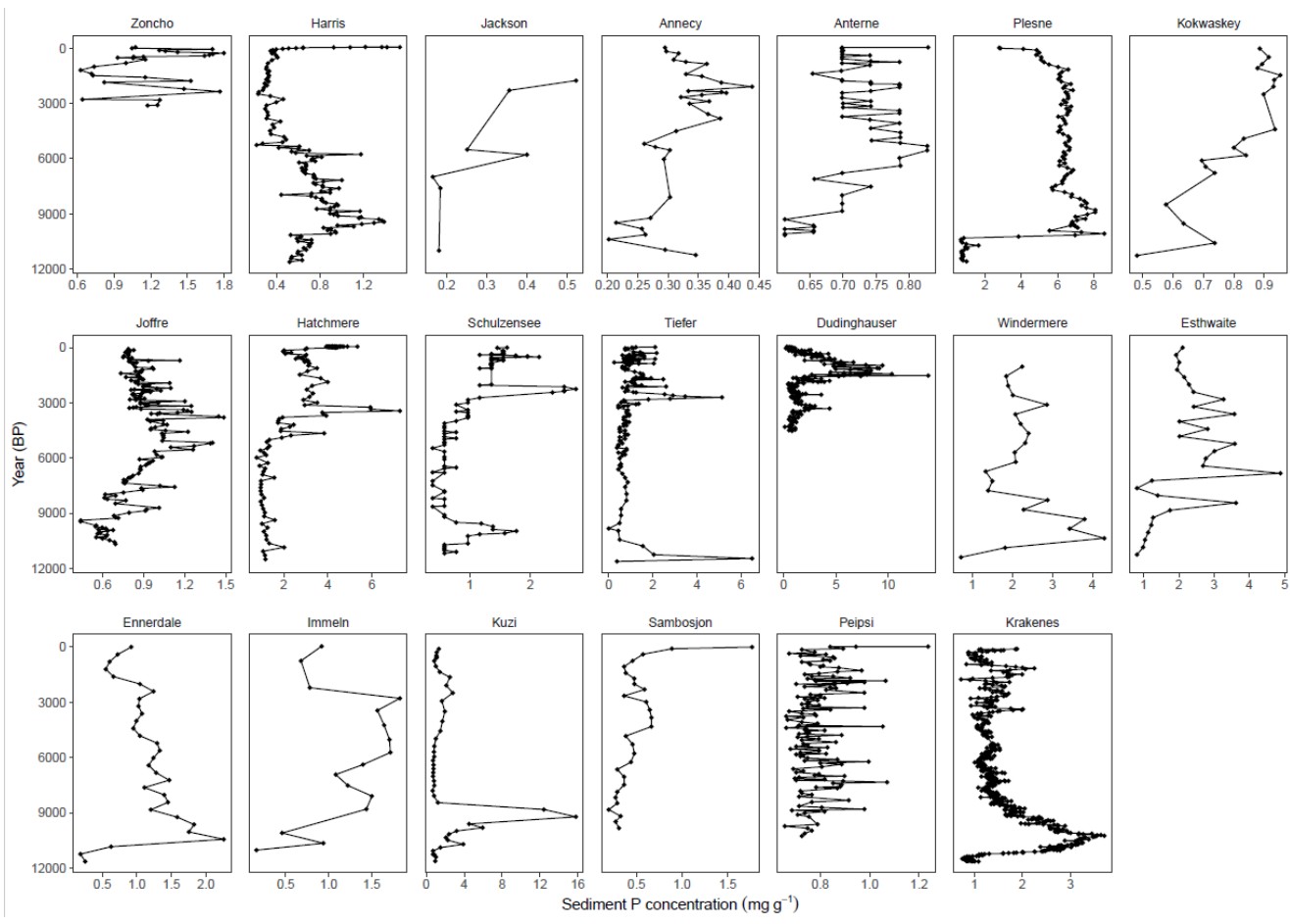

**Figure 3: The 20 sites with reported Holocene P concentration records**

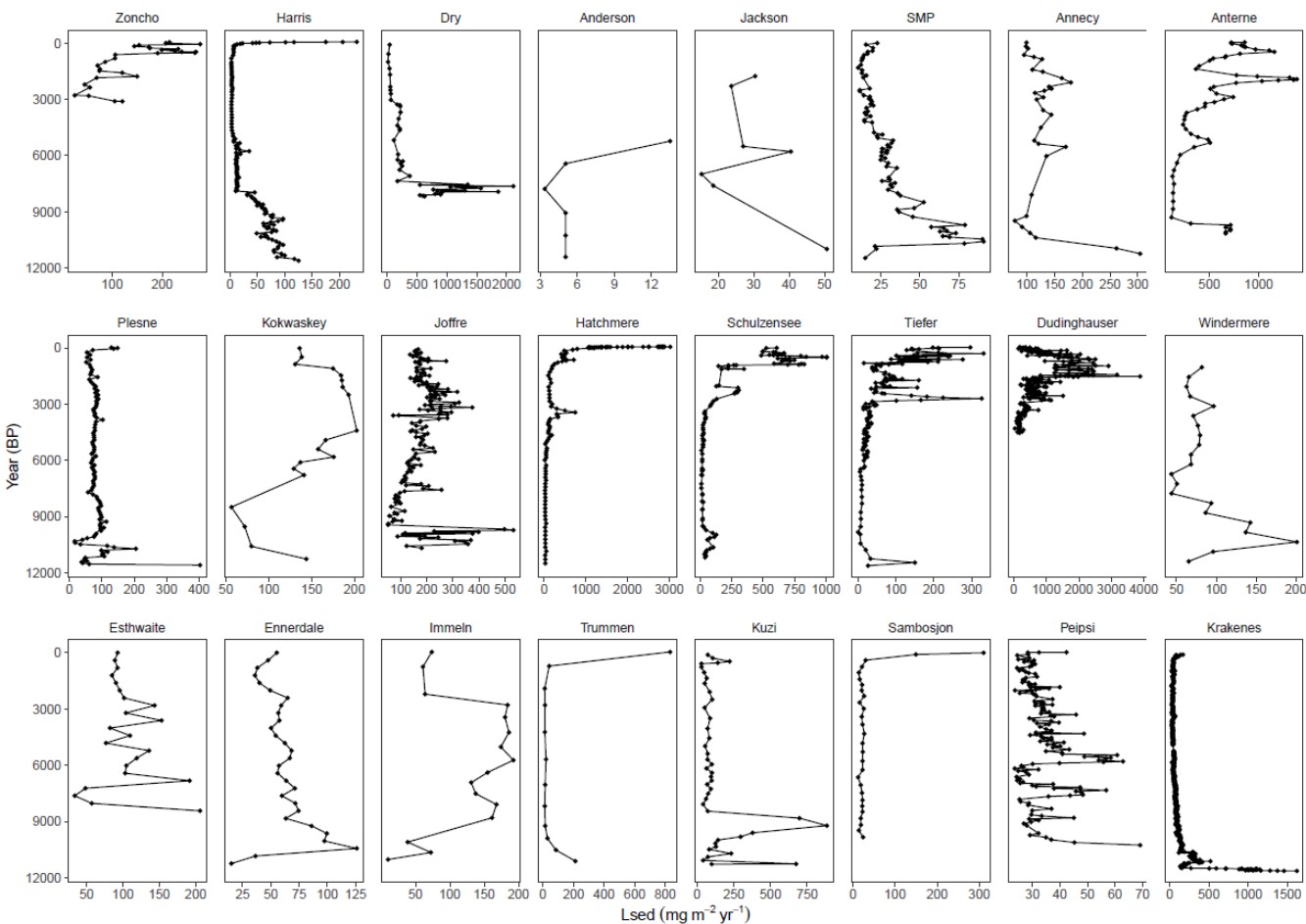

**Figure 4: The sediment inferred lake-wide sediment P loading ($L_{sed}$) records for all 24 lakes**

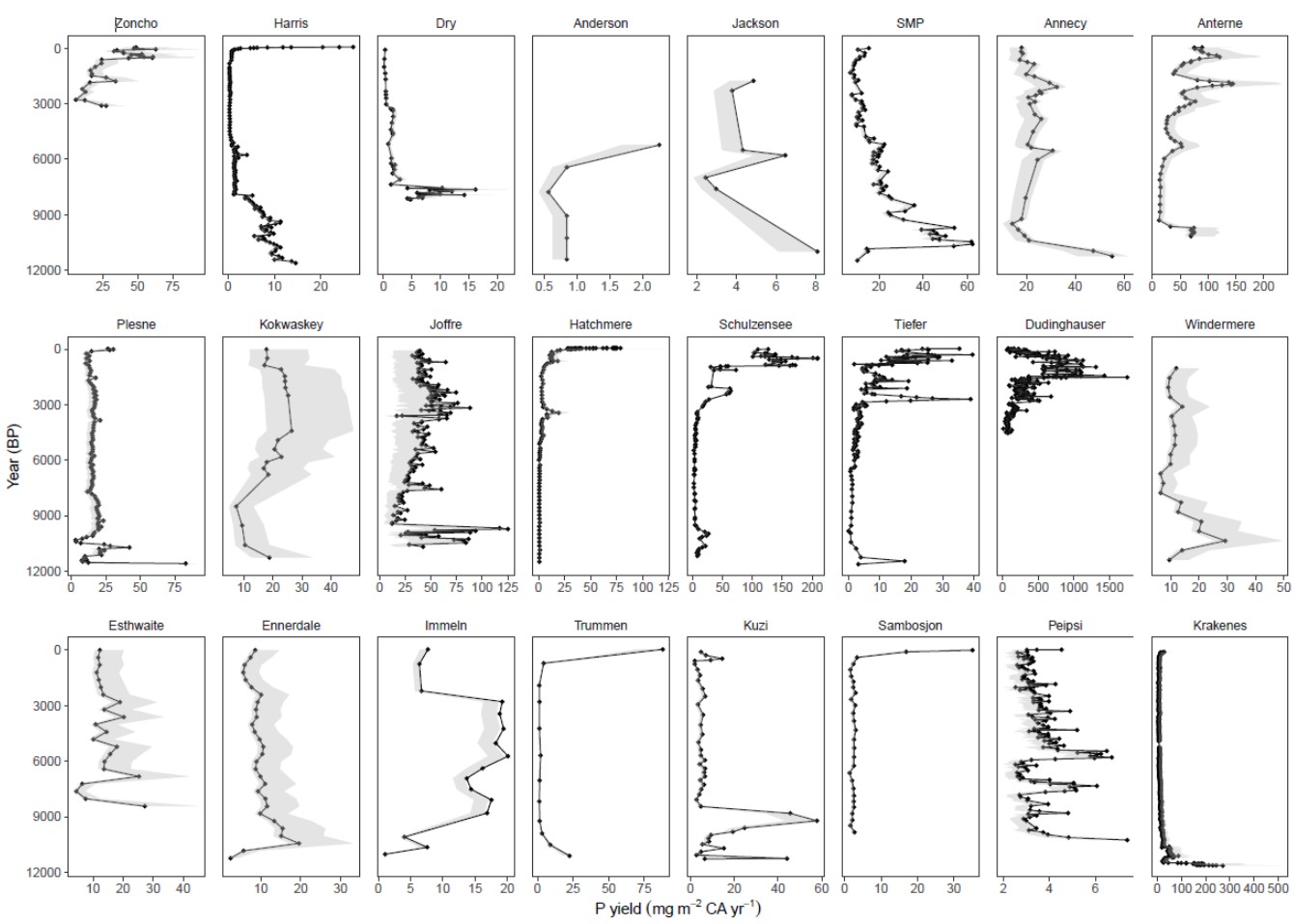

**Figure 5: The sediment inferred catchment P yield records for all 24 lakes**

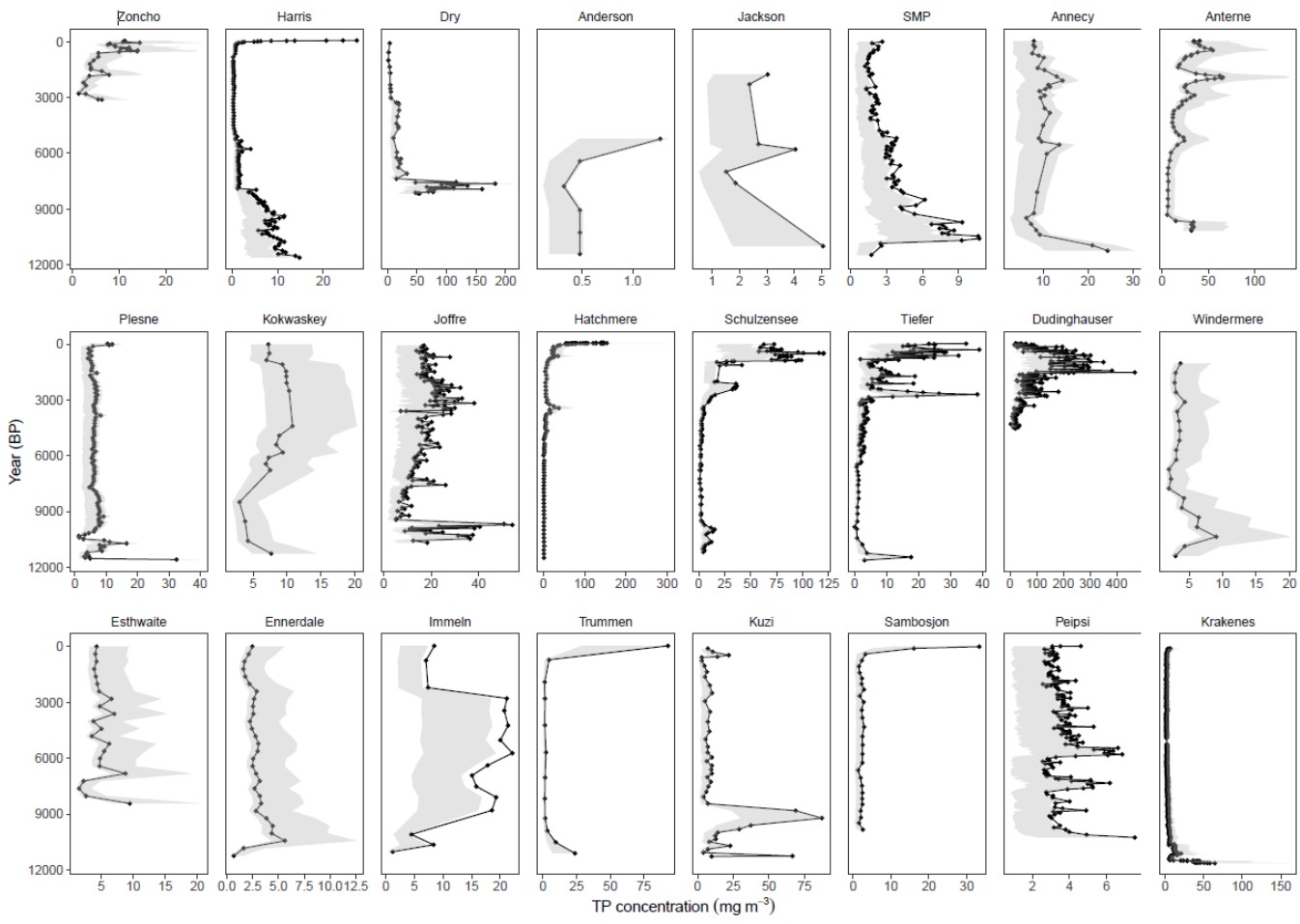

**Figure 6: The reconstructed sediment inferred lake water long-term mean TP (SI-TP) records for all 24 lakes**


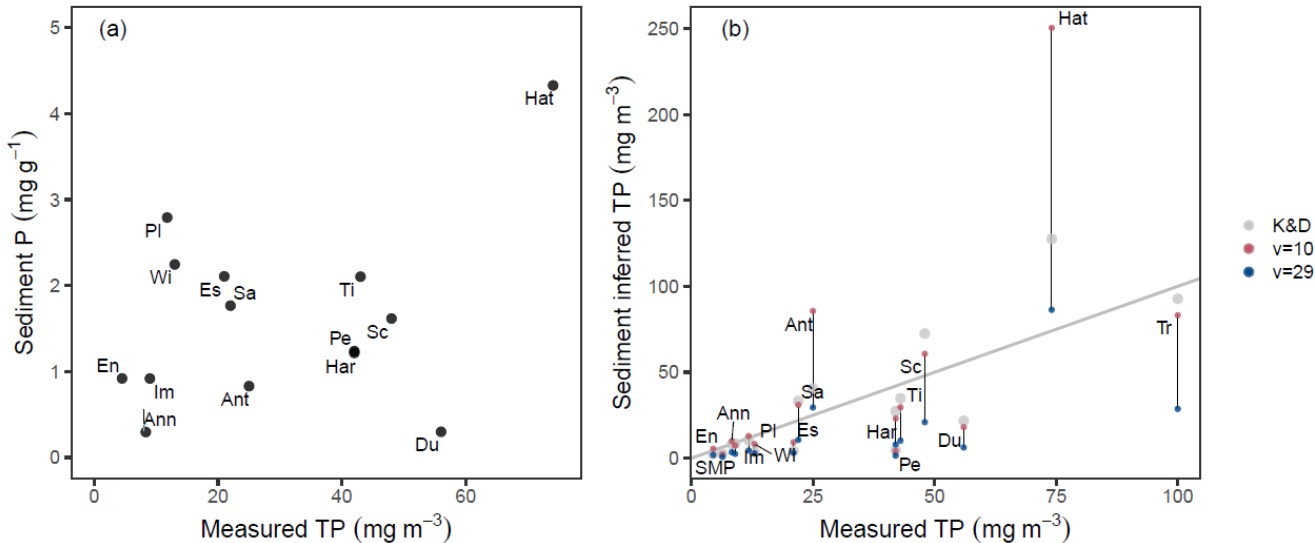

**Figure 7:** A comparison of measured lake water TP concentration with (a) sediment P concentration and (b) sediment inferred lake water TP for the 16 sites where lake water monitoring data exist. Three values of SI-TP are shown for each site which are calculated using the three $R_P$ values described in Sect. 2.1. The grey line shows the 1:1 relationship between TP and SI-TP. For site abbreviations see Fig. 1 caption

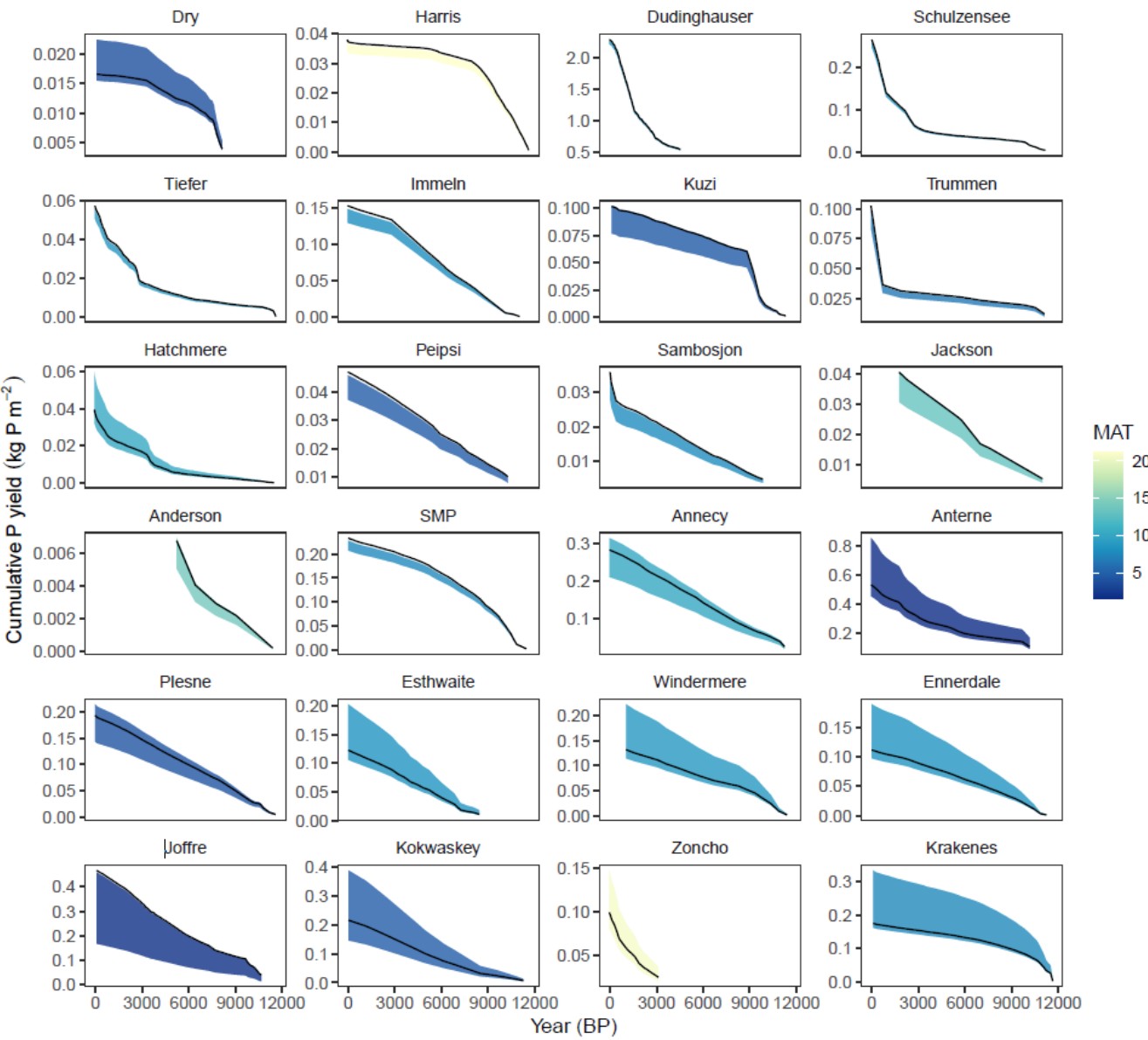


**Figure 8: Cumulative sediment inferred P yield for all 24 lakes, shaded by mean annual temperature and ordered by mean annual runoff (low to high)**

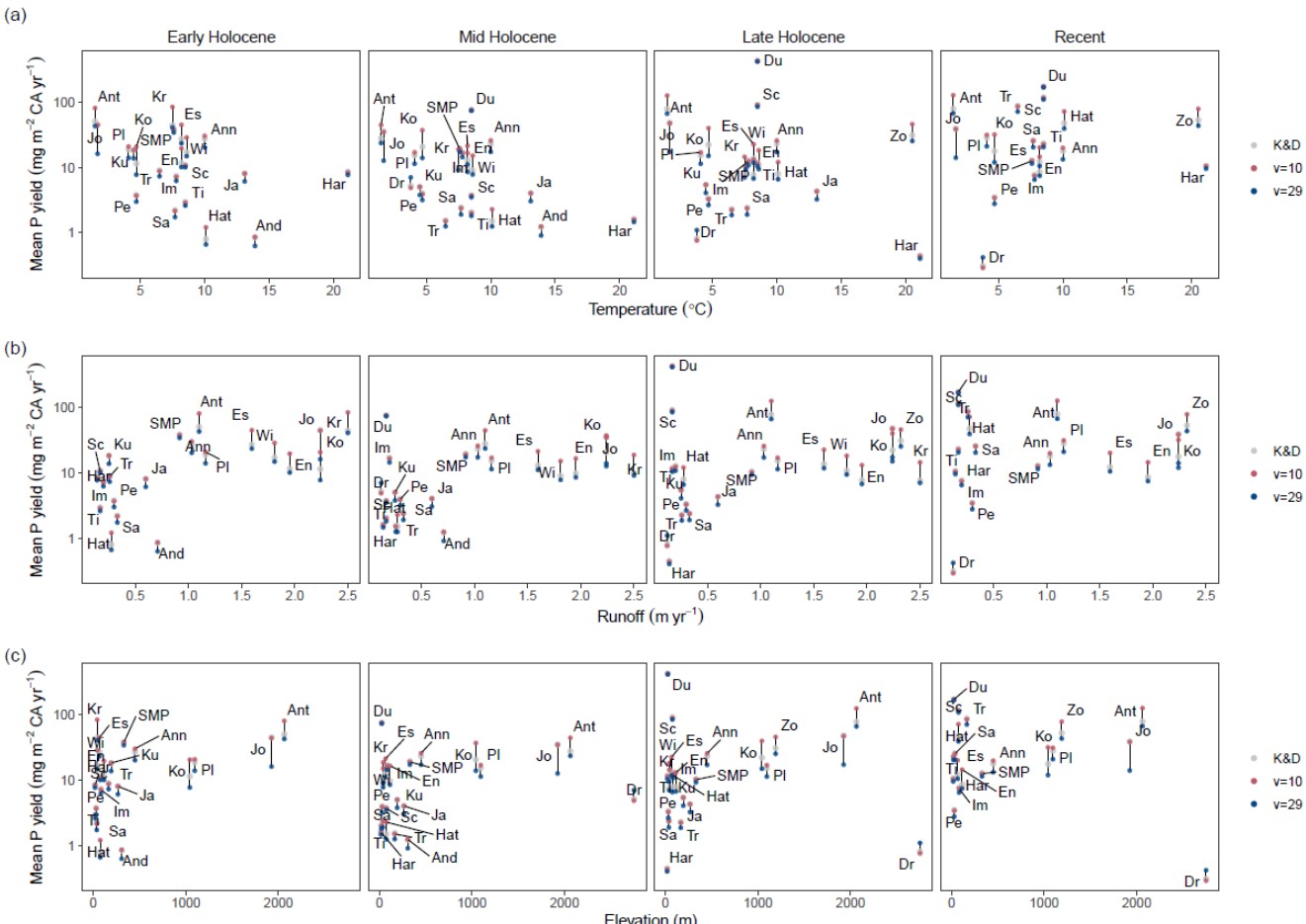

**Figure 9: Association of mean sediment inferred P yield for each site with (a) modern mean annual temperature, (b) modern mean annual runoff, and (c) lake elevation for the four Holocene time intervals. Mean sediment inferred P yield is estimated using three values of $R_p$. For site abbreviations see Fig. 1 caption and for corelations see Tables 1 and 2**

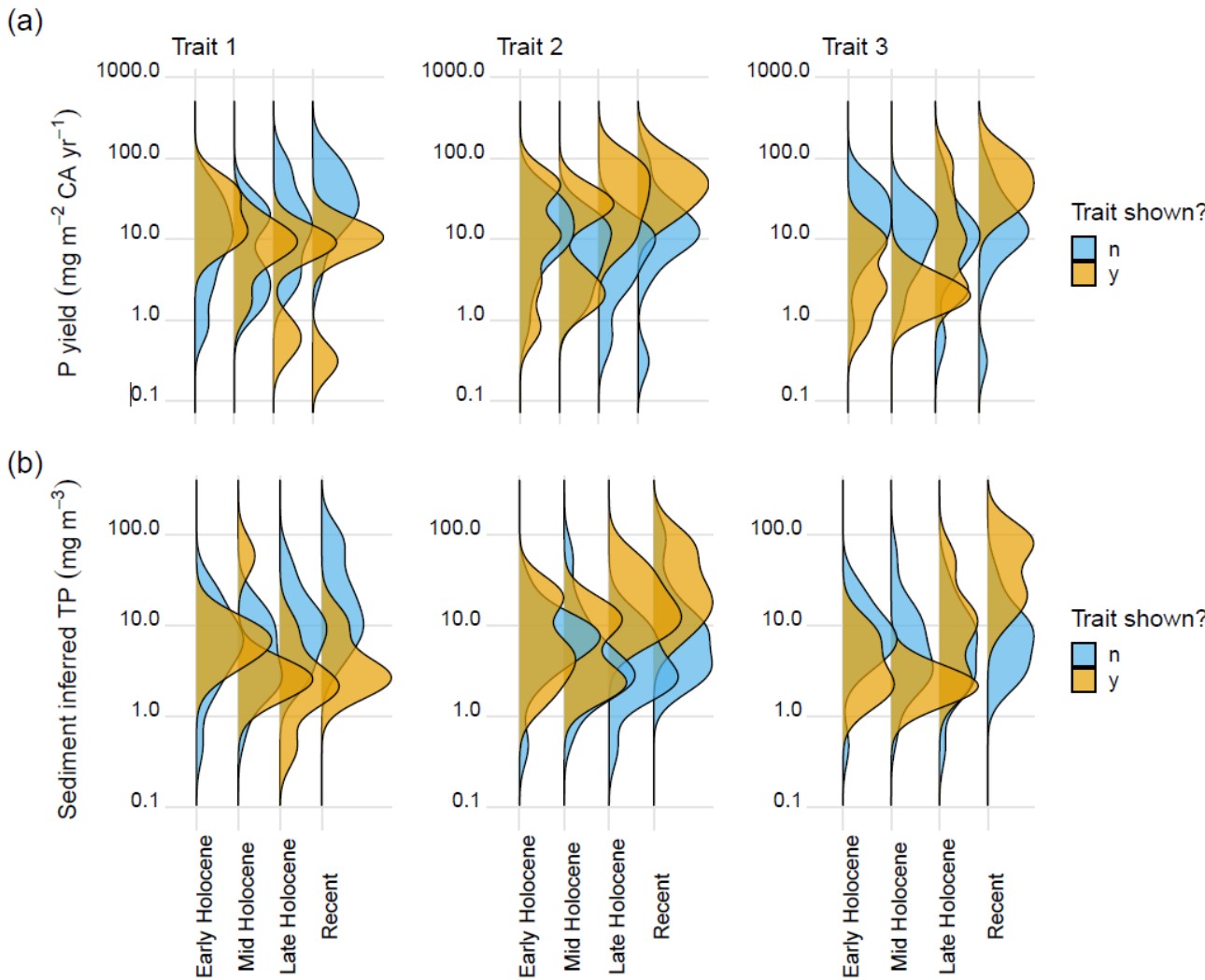

**Figure 10: Density distribution curves of (a) sediment mean P yield and (b) sediment inferred mean lake water TP showing changes in the three traits throughout the Holocene. Note Dudinghauser was removed for this analysis (see Sect. 3)**



**Table 1: P yield modelled on R (depth equivalent modern mean annual runoff; m yr⁻¹) and MAT (modern mean annual temperature; °C). See Figure 9 for graphs**

| | Adj. $R^2$ | F | Regression p | R coeff. | p | MAT coeff. | p |
|---|---|---|---|---|---|---|---|
| early Holocene | **0.33** | **5.91** | **0.011** | **0.52** | **0.043** | -0.65 | 0.108 |
| mid Holocene | **0.57** | **14.7** | **<0.001** | **0.56** | **0.002** | **-0.75** | **0.009** |
| late Holocene | **0.25** | **4.48** | **0.025** | **0.58** | **0.024** | -0.52 | 0.202 |
| Recent | - | | | | | | |

Note: Dudinghauser is excluded from this analysis


**Table 2: TP modelled on R (depth equivalent modern mean annual runoff; m yr⁻¹) and MAT (modern mean annual temperature; °C). See Figure 9 for graphs**

| | Adj. $R^2$ | F | Regression p | R coeff. | p | MAT coeff. | p |
|---|---|---|---|---|---|---|---|
| early Holocene | **0.16** | **7.41** | **0.014** | | | **-0.80** | **0.014** |
| mid Holocene | **0.39** | **14.6** | **0.001** | | | **-1.06** | **0.001** |
| late Holocene | **0.20** | **6.30** | **0.021** | | | **-0.84** | **0.021** |
| Recent | - | | | | | | |

Note: Dudinghauser is excluded from this analysis

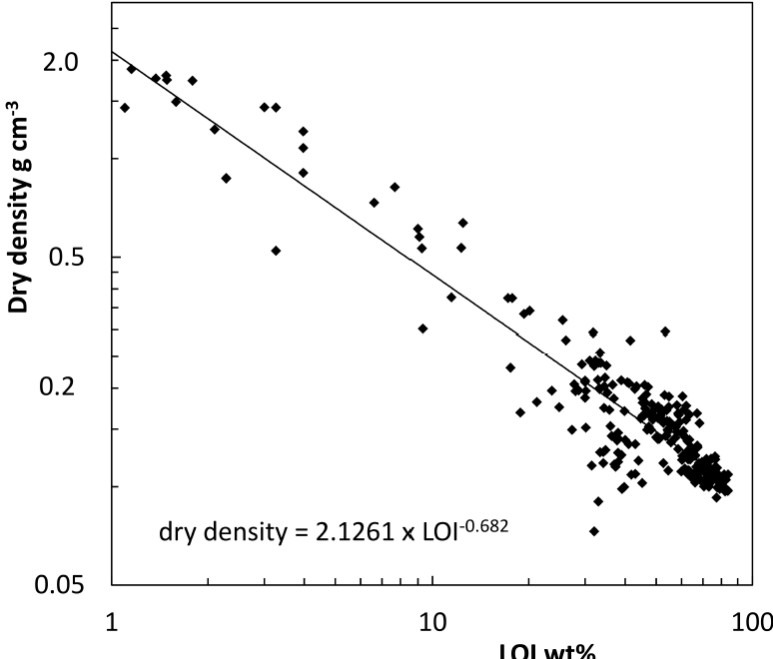

**Figure A 1: An empirical association between dry density and LOI is used to estimate dry density where neither dry density nor water contents were included. Data sources are described in Sect. 2.3.1**


**Table B 1: Parameters used to apply the model. The abbreviations used are: lake area ($A_L$); catchment area ($A_C$); lake volume (V); water residence time (T); water depth at coring site ($z_{core}$); mean lake water depth ($z_{mean}$); maximum lake water depth ($z_{max}$), depth equivalent runoff (R); mean annual temperature (MAT); mean annual precipitation (MAP)**

| Lake | Parameter | Source |
|---|---|---|
| Annecy | $A_C, A_L, V, T$ | Perga et al., 2015 |
| | $z_{core}$ | Loizeau et al., 2001 |
| Anterne | $A_C, A_L, z_{core}$ | Giguet-Covex et al., 2011 |
| | $V, T$ | Sesiano, 1993 |
| Tiefer | $A_C, A_L, z_{mean}, z_{max}$ | Selig et al., 2007 |
| | $MAP, MAT$ | climate-data.org, 2021 (Reanalysis) |
| Schulzensee | $A_C, A_L, z_{mean}, z_{max}$ | Selig et al., 2007 |
| | $MAP, MAT$ | climate-data.org, 2021 (Reanalysis) |
| Dudinghauser | $A_C, A_L, z_{mean}, z_{max}$ | Selig et al., 2007 |
| | $MAP, MAT$ | climate-data.org, 2021 (Reanalysis) |
| Ķuži | $A_C, A_L, z_{mean}, z_{max}, V, T$ | Terasmaa et al., 2013 |

| | | |
|---|---|---|
| Ennerdale | $A_C, A_L, z_{mean}, z_{max}, V, T$ | Ramsbottom, 1976 |
| | $R$ | NRFA, 2021 (long-term river flow data) |
| Esthwaite | $A_C, A_L, z_{mean}, z_{max}, V, T$ | Ramsbottom, 1976 |
| | $R$ | NRFA, 2021 (long-term river flow data) |
| Windermere | $A_C, A_L, z_{mean}, z_{max}, V, T$ | Ramsbottom, 1976 |
| | $R$ | NRFA, 2021 (long-term river flow data) |
| Peipsi | $A_C, A_L, z_{mean}, z_{core}$ | Kisand et al., 2017 |
| | $Q$ | Stålnacke et al., 1998 |
| Plesne | $A_C, A_L, V$ | Kopáček et al., 2004 |
| | $T, R$ | Kopáček et al., 2006 |
| | $z\text{-}core$ | Norton et al., 2016b |
| Hatchmere | $A_C, A_L, z_{mean}, z_{core}, R$ | Boyle et al., 2015 |
| Sargent Mountain | $A_C, A_L, z_{core}$ | Norton et al., 2011 |
| | $MAP, MAT$ | Perry, 2007 |
| Kråkenes | $A_C, A_L,$ | Boyle et al., 2013b |
| | $MAP, MAT$ | seNorge, 2021 |
| Immeln | $A_C, A_L,$ | SMHI, 2013 |
| | $z_{core}, z_{mean}$ | Digerfeldt, 1974 |
| | $MAP, MAT$ | climate-data.org, 2021 (Kristianstad) |
| Trummen | $A_C, A_L, z_{mean}, z_{max}, T$ | World Lake Database, 2021 |
| Anderson | $A_L$ | Liu et al., 2013 |
| | $MAP, MAT$ | Climate-data.org (Sparta, TN) |
| Jackson | $A_L$ | Liu et al., 2013 |
| | $MAP, MAT$ | Climate-data.org (Elizabethtown, KT) |
| Kokwaskey | $A_C$ | Souch, 2004 |
| | $R$ | Menounos, 2002 |
| Lower Joffre | $A_C, A_L, z_{mean}, z_{core}, R$ | Menounos, 2002 |
| Harris | $A_L, z_{mean}$ | Kenney et al., 2016 |
| | $MAP, MAT$ | climate-data.org, 2021 (The Villages) |
| | $R$ | USGS 02238000 HAYNES CREEK AT |

| | | LISBON, FL U.S. Geological Survey, 2020 |
|---|---|---|
| Sämbosjön | $A_C$, $A_L$, $z_{core}$ | Digerfeldt and Håkansson, 1993 |
| | $MAP$, $MAT$ | climate-data.org, 2021 (Träslövsläge) |
| Dry Lake | $A_C$ | Bird and Kirby, 2006 |
| | $R$ | USGS 11051499 SANTA ANA R NR MENTONE (RIVER ONLY) CA, U.S. Geological Survey, 2020 |
| Laguna Zoncho | $A_C$, $A_L$ | Filippelli et al., 2010 |
| | $z_{core}$, $MAP$, $MAT$ | Clement and Horn, 2001 |


**Table B 2: Sources of data about the sediment records. MAR = mass accumulation rate and SAR = sediment accumulation rate**

| Lake | Record | Published units | Source |
|---|---|---|---|
| Annecy | P concentration | µg g$^{-1}$ | Loizeau et al., 2001 |
| | Water content | wt% | Loizeau et al., 2001 |
| Anterne | P concentration ($P_2O_5$) | wt% | Giguet-Covex et al., 2011 |
| | SAR | cm yr$^{-1}$ | Giguet-Covex et al., 2011 |
| Tiefer | P concentration | mg g$^{-1}$ | Selig et al., 2007 |
| | LOI | wt% | Selig et al., 2007 |
| Schulzensee | P concentration | mg g$^{-1}$ | Selig et al., 2007 |
| | LOI | wt% | Selig et al., 2007 |
| Dudinghauser | P concentration | mg g$^{-1}$ | Selig et al., 2007 |
| | LOI | wt% | Selig et al., 2007 |
| Ķuži | P concentration | mg g$^{-1}$ | Terasmaa et al., 2013 |
| | MAR | g m$^{-2}$ yr$^{-1}$ | Terasmaa et al., 2013 |
| Ennerdale | P concentration | µg g$^{-1}$ | Mackereth, 1966 |
| | LOI | wt% | Mackereth, 1966 |
| Esthwaite | P concentration | µg g$^{-1}$ | Mackereth, 1966 |

| | | | |
|---|---|---|---|
| | LOI | wt% | Mackereth, 1966 |
| Windermere | P concentration | $\mu g\ g^{-1}$ | Mackereth, 1966 |
| | LOI | wt% | Mackereth, 1966 |
| Peipsi | P concentration | $\mu g\ g^{-1}$ | Kisand et al., 2017 |
| | LOI | wt% | Leeben et al., 2010 |
| Plesne | P concentration | $mg\ g^{-1}$ | Kopacek (pers. comm)/ Kopáček et al., 2007 |
| | MAR | $g\ m^{-2}\ yr^{-1}$ | Kopacek (pers. comm)/ Norton et al., 2016b |
| Hatchmere | P concentration | $mg\ g^{-1}$ | Boyle et al., 2015 |
| Sargent Mountain | P flux | $\mu mol\ cm^{-2}\ yr^{-1}$ | Norton et al., 2011 |
| Kråkenes | P concentration | $mg\ g^{-1}$ | Boyle et al., 2013b |
| Immeln | P concentration | $mg\ dm^{-3}$ | Digerfeldt, 1974 |
| | SAR | $mm\ yr^{-1}$ | Digerfeldt, 1974 |
| | Dry density | $g\ dm^{-3}$ | Digerfeldt, 1974 |
| Trummen | P flux | $g\ m^{-2}\ yr^{-1}$ | Digerfeldt, 1972 |
| Jackson Pond | P flux | $\mu mol\ cm^{-2}\ kyr^{-1}$ | Filippelli and Souch, 1999 |
| Anderson Pond | P flux | $\mu mol\ cm^{-2}\ kyr^{-1}$ | Filippelli and Souch, 1999 |
| Kokwaskey | P flux | $\mu mol\ cm^{-2}\ kyr^{-1}$ | Filippelli and Souch, 1999 |
| Lower Joffre | P concentration | $\mu mol\ g^{-1}$ | Filippelli et al., 2006 |
| | Dry density | $g\ cm^{-3}$ | Menounos, 2002 |
| Harris | P concentration | $mg\ g^{-1}$ | Kenney et al., 2016 |
| | Water content | wt% | Kenney et al., 2016 |
| Sämbosjön | P flux | $g\ m^{-2}\ yr^{-1}$ | Digerfeldt and Håkansson, 1993 |
| Dry Lake | P flux | $\mu mol\ cm^{-2}\ kyr^{-1}$ | Filippelli and Souch, 1999 |
| Laguna Zoncho | P concentration | $\mu mol\ g^{-1}$ | Filippelli et al., 2010 |
| | Inorganic content | wt% | Filippelli et al., 2010 |


Table B 3: rbacon (Blaauw and Christen, 2011) settings for cores where new age models were developed

| Lake | Sections | Thickness | Acc.shape | Acc.mean | Mem.strength | Mem.mean |
|---|---|---|---|---|---|---|
| Annecy | 101 | 10 | 1.3 | 10 | 8 | 0.45 |
| Tiefer | 198 | 5 | 1.5 | 10 | 10 | 0.2 |
| Schulzensee | 94 | 15 | 1.3 | 10 | 8 | 0.45 |
| Dudinghauser | 107 | 10 | 1.3 | 5 | 8 | 0.45 |
| Peipsi | 74 | 5 | 1.5 | 20 | 6 | 0.2 |
| Immeln | 67 | 10 | 1.3 | 10 | 8 | 0.45 |
| Trummen | 58 | 10 | 1.3 | 20 | 8 | 0.45 |
| Lower Joffre | 60 | 10 | 1.3 | 20 | 8 | 0.45 |
| Harris | 60 | 10 | 1.3 | 20 | 8 | 0.45 |
| Laguna Zoncho | 30 | 10 | 1.3 | 10 | 8 | 0.45 |


Table B 4: Site data are from the sources listed in Table B1. The population density data are approximations derived by summing the present-day population totals for settlements within the catchment

| Lake name | Name abbrev. | Latitude decimal ° | Longitude decimal ° | Elev. m | $A_L$ km$^2$ | $A_C : A_L$ | R m yr$^{-1}$ | MAT °C | Water loading m yr$^{-1}$ | Pop. density km$^{-2}$ | TP mg m$^{-3}$ |
|---|---|---|---|---|---|---|---|---|---|---|---|
| Lac D'Annecy | Ann | 45.86 | 6.17 | 450 | 27 | 10.1 | 1.03 | 10 | 10.4 | 160 | 8 |
| Plesne Lake | Pl | 48.78 | 13.87 | 1095 | 0.075 | 8.9 | 1.16 | 4.1 | 10.3 | 0 | 12 |
| Hatchmere | Hat | 53.25 | -2.67 | 74 | 0.0345 | 82.3 | 0.274 | 10.1 | 22.6 | 70 | 74 |
| Peipsi | Pe | 58.65 | 27.46 | 30 | 3555 | 13.4 | 0.299 | 4.7 | 4.0 | 21 | 42 |
| Sargent Mountain Pond | SMP | 44.33 | -68.27 | 329 | 0.0075 | 1.7 | 0.916 | 7.6 | 1.6 | 0 | 6 |
| Dudinghauser | Du | 53.91 | 12.21 | 25 | 0.188 | 2.3 | 0.168 | 8.5 | 0.4 | 18 | 56 |
| Schulzensee | Sc | 53.29 | 12.80 | 75 | 0.485 | 5.3 | 0.168 | 8.5 | 0.9 | 0 | 48 |
| Tiefer | Ti | 53.79 | 12.29 | 21 | 0.159 | 10.1 | 0.168 | 8.5 | 1.7 | 0 | 43 |
| Lac D'Anterne | Ant | 45.99 | 6.80 | 2063 | 0.12 | 21.3 | 1.1 | 1.5 | 25.7 | 0 | 25 |
| Lake Harris | Har | 28.78 | -81.80 | 18 | 75 | 10.0 | 0.14 | 21.1 | 1.4 | 142 | 42 |
| Jackson Pond | Ja | 37.43 | -85.73 | 265 | 0.035 | 10.0 | 0.6 | 13.1 | 1.2 | 2 | n/a |
| Anderson Pond | And | 36.03 | -85.50 | 306 | 0.13 | 10.0 | 0.71 | 13.9 | 1.4 | 32 | n/a |
| Dry Lake | Dr | 34.12 | -116.83 | 2750 | 0.05 | 74.0 | 0.12 | 3.8 | 8.9 | 0 | n/a |
| Kokwaskey Lake | Ko | 50.12 | -121.83 | 1042 | 0.46 | 91.3 | 2.24 | 4.7 | 204.5 | 0 | n/a |
| Windermere | Wi | 54.34 | -2.94 | 40 | 14.76 | 15.6 | 1.81 | 8.6 | 28.3 | 75 | 13 |

| Ennerdale | En | 54.52 | -3.38 | 114 | 2.999 | 14.7 | 1.953 | 8.2 | 28.7 | 1 | 5 |
| Esthwaite | Es | 54.36 | -2.99 | 66 | 1.004 | 17.0 | 1.596 | 8.2 | 27.2 | 30 | 21 |
| Kråkenesvatn | Kr | 62.03 | 5.00 | 41 | 0.055 | 14.8 | 2.5 | 7.5 | 37.0 | 27 | n/a |
| Laguna Zoncho | Zo | 8.81 | -82.96 | 1190 | 0.75 | 9.3 | 2.32 | 20.5 | 21.7 | 0 | n/a |
| Lower Joffre Lake | Jo | 50.37 | -122.50 | 1925 | 0.104 | 139.4 | 2.24 | 1.7 | 312.3 | 0 | n/a |
| Sämbosjön | Sa | 57.16 | 12.42 | 37 | 0.23 | 12.9 | 0.33 | 7.7 | 4.3 | 4 | 22 |
| Trummen | Tr | 56.86 | 14.83 | 165 | 1 | 13.0 | 0.258 | 6.5 | 3.4 | 1000 | 100 |
| Immeln | Im | 56.27 | 14.33 | 81 | 24 | 12.3 | 0.199 | 7.8 | 2.4 | 13 | 9 |
| Ķuži | Ku | 57.03 | 25.33 | 192 | 0.063 | 24.6 | 0.251 | 4.5 | 6.2 | 2 | n/a |


**Table B 5: Recent TP data for comparison with model inferred TP**

| Lake | TP (mg m$^{-3}$) | Year(s) measured | Reference |
| --- | --- | --- | --- |
| Annecy | 8.33 | 1971-2006 | Perga et al., 2010 |
| Plesne | 11.78 | 2000 | Kopáček et al., 2006 |
| Hatchmere | 74.09 | 2000-2015 | EA, 2021 |
| Peipsi | 42 | ? | Nõges et al., 2007 |
| SMP | 6.4 | ? | Perry, 2007 |
| Dudinghauser | 56 | 1999-2000 | Selig et al., 2007 |
| Schulzensee | 48 | 1999-2000 | Selig et al., 2007 |
| Tiefer | 43 | 1999-2000 | Selig et al., 2007 |
| Anterne | 25 | 1998-2018 | Rimet et al., 2020[1] |
| Harris | 42 | 1980-1990 | Fulton and S, 1995 |
| Windermere | 13 | 2010 | Maberly and Elliott, 2012 |
| Ennerdale | 4.5 | 2015-2019 | EA, 2021 |
| Esthwaite | 21 | 2010 | Maberly and Elliott, 2012 |
| Sämbosjön | 22 | ? | Digerfeldt and Håkansson, 1993 |
| Trummen | 100 | 1968-1978 | Cronberg, 1982 |
| Immeln | 9 | 1997 | Berglund et al., 2001 |

[1] Data from sampling point 1, corresponding to the deepest and pelagic part of the lake. Data were obtained bi-monthly,
except for the winter period, for which sampling is performed once a month data (© OLA-IS, AnaEE-France, INRAE of
Thonon-les-Bains, Asters, GIS LACS SENTINELLES, https://doi.org/10.4081/jlimnol.2020.1944)