# Peer review of "Towards a history of Holocene P dynamics for the northern hemisphere using lake sediment geochemical records"

_Biogeosciences, 2021_

## Referee Comment (RC2)

**General comments**

Many northern lakes have experienced nutrient enrichment over recent decades to centuries due to either N deposition or P inputs or both, with resulting changes in chemical and biological conditions in many of those lakes. The extent to which most lakes have changed is difficult to establish given that contemporary monitoring programs were initiated after inputs of P (and N) had already long since changed, along with many other biogeochemical and physical parameters. To identify the scale of potential changes in lake conditions we require estimates of what lake conditions may have been prior to disturbance, and with this information scientifically motivated remediation/restoration or management goals can be developed. The analysis of lake sediment records has long since been an established as a means to address these questions.

As noted by Moyle et al. in this manuscript and in a new paper by Moyle and Boyle (2021), which lays the foundation for this study, the main method to establish reference values for past lake-water P and how conditions may have changed in response to natural environmental and climate changes or human-driven impacts has been application of diatom-inferred models. While the accuracy of the underlying transfer functions can be questioned (Juggins 2013), the fundamental inferred changes in nutrient conditions are robust. However, as Moyle et al. observe, there are more geochemical records available, which include data on sediment P, than diatom records, and if it is possible to exploit these existing geochemical data sets then it would be possible to greatly increase the spatial and temporal estimates of lake-water P changes and also landscape P yield in response to a wide range of natural or human-driven changes. Certainly it is possible to discuss whether total P concentrations (along with sediment accumulation) are sufficient, or whether other geochemical data would be ideal to include to somehow partition the P into different fractions that may represent different sources of P to the sediment record (e.g., using PCA or actual speciation extractions such as non-apatite inorganic P). This latter is outside the scope of their study.

The sediment-inferred lake-water TP model is already presented in Moyle and Boyle, so I have no reason to try to dig through that foundational work. That study also included a limited comparison of their modeled values against available monitoring data, where there is sufficient overlap between their sediment reconstruction and monitoring data in three of the six lakes to indicate the sediment-inferred TP lines up with the temporal pattern, which would at least suggest there is some basic credence to the model. But it is unfortunate the sediment records in, for example, Lakes Erie and Ontario do not extend to more recent years, where monitoring shows a distinct decline in P. It would have been most ideal in that study to have a few records with more-extensive overlap between the sediment model and monitoring.

In the new Moyle and Boyle paper, the authors included one comparison of their sediment-inferred TP with diatom-inferred TP for a N Ireland lake record for the period 1850–1990's, which suggests overall similar patterns; however, the timing of a more rapid increase differs between the two modeling approaches. If one considers that the overall long-term pattern is of greatest value, then this may be acceptable, but the discrepancy in the temporal pattern of the increase is notable. As well as the desirability of a more extensive overlap between the sediment model and monitoring data, it would have been useful to test a few more

sediment versus diatom reconstructions. With that in mind, that is the one thing missing in this new manuscript — a few comparisons of the sediment-inferred lake-water TP versus diatom-inferred TP or at least inferred changes in nutrient levels if TP itself was not modeled from the diatom data.

Overall, although modeling-focused papers by nature are notoriously not the easiest or most exciting to read, the manuscript is well written despite the technical nature of some sections and is logically structured. And broadly I think there is good merit in the approach to exploit geochemical records to estimate past lake-water TP concentrations, which is not without its challenges as the authors do discuss. Consequently, the current text itself is good; however, what I feel is missing and would make the evaluation of this approach more robust and of wider value would be more specific comparisons of the sediment-inferred lake-water TP with available diatom-inferred reconstructions, which is the current tool for estimating Holocene timescale lake-water TP concentrations. Diatom-inferred TP reconstructions do exist for some of the lakes, such as Sargent Mountain Pond (Norton et al.) and the most recent c. 1500 years in the three lowland German lakes (Hübener et al.) – off-hand I do not know about data for other lakes; for Kråkenes there are data at least for the early Holocene, but would guess more complete diatom/chironomid data ought to exist – or at least inferences of nutrient changes that could be inferred from diatom data (without specifically having diatom-inferred TP). Not that diatom-inferred TP represents necessarily the true value, but it is the established approach for inferring Holocene patterns in lake-water TP and represents at present the only way to 'validate' the sediment model. The authors also might at least consider if there is any merit to comparing their sediment-inferred TP patterns with sediment P speciation, which exist for Sargent Mountain Pond (Norton et al.) and Anterne (Giguet-Covex et al. GCA 2013). This suggestion represents mainly an addition to the section 4.5 Reliability and Limitations., but is a significant enough addition that would correspond to a major revision even if the main text is fine.

**Some specific comments**
Numbers refer to line number
- General curiosity question as to why P water concentrations are reported here as mg/m3 rather than µg/L?
- 33: odd expression to write "recovery from glaciation". Maybe use landscape development following deglaciation?
- 36: for clarity I prefer avoiding use of 'as' as synonym for 'because'. (Amongst other writing guidelines, I am particularly a fan of the USGS 'Suggestions to Authors of the Reports of the United States Geological Survey', but should note this is not an American/British English difference; rather, it makes for greater clarity.)
- 47: concentrations
- 81: Lakes, which …
- 123: it is not unique to apply a value of 2.7 g/cm3 for the density of the non-LOI (ash) fraction of the sediment (comparable to the density of basalt); however, how does this account for the fact that much of the ash fraction is composed of diatoms? From a quick search I came up with one value from Sañé et al. (PLOSone 2013) of 2.0. Also, in boreal lakes how much of the, e.g., Fe – another major constituent – is mineral matter rather than in organic complexes?

- 206: Sargent Mountain Pond. Norton et al. had also included a diatom-inferred TP reconstruction and it would be very valuable in this manuscript to compare and contrast the values derived from the Moyle-Boyle (2021) model with that derived from diatoms. While the approximate average Holocene values might be relatively similar, I would not consider the Holocene pattern to be all that similar. The geochemistry model indicates a peak in lake-water TP of about 9 μg/L during c. 10000–11000 BP, with a more rapid decline until c. 8000 and thereafter a steady, long-term decline until c. 1000 BP. However, in Norton et al. diatom-inferred TP is about 5 μg/L during c. 13000–3000 BP (key to note the reconstruction is low resolution), with slightly higher values c. 7000 BP. This would be important to discuss, as well as if similar data exist for any of the other sediment records.
- 271: Temporal divisions. It is practical to divide the records into established periods, so no fundamental issue with that; however, I would consider it a preconception to assume (lines 276-277) that climatic changes are "likely to have impacted the P records", because as later noted for Trait 2 many of the lakes have been impacted since the mid-Holocene, as well as that most of the lakes really do not show any changes across the major climatic periods.
- 408–: Anterne: Giguet-Covex et al. do discuss neoglaciation as a possible factor for the increase in sedimentation at 5500 BP, but they also acknowledge that human activities may also have been involved, given the presence of anthropogenic indicators from 5660 BP. Arnaud and co-workers in other works have suggested that in some of the alpine lakes at least that human activities destabilized catchment soils, which in turn made the catchments more sensitive to climate (and hence hydrological changes were subsequently recorded in the sediment records, but less so prior). Furthermore, the changes in sediment during 4600–2400 BP Giguet-Covex et al. also attribute largely to land use, where 'erodibility' rather than increased discharge was the main driver; notably, glacial activity was not discussed.
  With the Anterne data it would also be interesting how the sediment-inferred lake-water TP compares more specifically to other aspects of the sediment record, which would include also the XANES and P speciation from this record by Giguet-Covex et al. (GCA 2013). One concern is that the sediment-inferred lake-water TP more-or-less reproduces the sedimentation rate.
- 444: … exhibiting **the** stable phase seen …
- 454: … the total P supply **to** each of the **lakes**.
- 619: north German lakes: a comparison is made of the average inferred TP concentration for the past 1000 years between the sediment-inferred model and diatom-inferred reconstruction. However, here is an excellent opportunity to compare patterns between the two models over the 1500 years reported for the diatom-inferred TP reconstructions for these three lakes. Based on visual comparison, there seems to be substantial discrepancy between the sediment- and diatom-inferred reconstructions with a very sharp increase in the sediment-inferred TP c. 1000 BP that is not apparent in the diatom-inferred TP, which does not show a major change until the past century (figures pasted in below). Moyle et al. mention the discrepancy in the values for Dudinghauser, but do not delve into the other records. Addressing the comparison in records with available diatom data (whether inferred TP or just inferring nutrient changes) would be a very valuable addition to the manuscript.

[Figure]

**Tiefer**

---

## Author Response (AR1)

Line numbers refer to the original document

**Changes after reviewer comments - Gabriel Filippelli**

Lines 293/4 and 370 – R2 values have been added to support the statement "SI-TP correlates with measured lake water TP"

Section 2.1 – A statement has been added to clarify that P yield is not calculated from runoff "This means that the calculated P yield values come directly from the sediment record and not from a hydrochemical mass balance, i.e. runoff and inflow TP concentration."

Section 2.3.3 – Individual methodologies for calculating runoff are now included

Line 349 – we have now made it clearer that runoff refers to modern mean annual runoff

Also line 461 and 465, Fig 9, and Tables 1 and 2 captions – we have made it clear that temperature and runoff refer to modern mean annual values

Section 4.5 – we have added a paragraph commenting on the use of a fixed focussing factor. This paragraph also comments why a variable value for runoff is not used and comments on the resulting impacts of using fixed Rp and qs values and their effect on the model outcome.

The Figure 7b caption has been amended to include a reference to the 1:1 line and a clarification on the calculation of the SI-TP values

The Figure 9 caption now cross refers to Tables 1 and 2 and vice versa

**Changes after reviewer comments - Richard Bindler**

Lines 7 and 33 – changed "recovery from glaciation" to "landscape development following glaciation"

Line 36, 207, 212, 219, 220, 228, 233, 238, 253, 259, 262, 270, 388, 403, 619 – changed "as" to "because"

Line 47 – concentrations

Line 81 – Lakes, which

Section 2.2 – at the end of the section we have commented on our use of mg/m3 over ug/L

Line 276/277 - The sentence containing the preconception about climatic impacts on the sediment P record has been removed

Line 408 onwards – we have clarified that neoglaciaton refers to regional neoglaciation

Line 444 – exhibiting the stable phase seen…

Line 545 – …the total P supply to each of the lakes.

Section 4.5 – we have expanded the paragraph comparing SI-TP to DI-TP to include observational comparisons of the profiles of the three German lakes and have added an observational comparison with the diatom record from Sargent Mountain Pond. We have also included a brief interpretation of the differences in the records.

**Author changes**

Abstract – Rephrased the reference to comparison of SI-TP with DI-TP

Line 338 – Figure reference changed from Fig. 07 to refer to correct figure (Fig. 08)

Table B1 – added missing data from Jackson pond and Anderson Pond

Table B4 caption – table reference changed from SM1 to B1

To avoid confusion, we have made sure the Moyle and Boyle (2021) model is consistently referred to as the "SI-TP model" throughout (rather than "the model" or "the phosphorus model" etc as previously)

---

## Author Response (AR2)

We have experimented with combining Figure 3-6 but cannot find a combination that does not result in unacceptable loss of information. As each figure contains its own specific information, we argue that these should be retained in the paper and as a result have not made any changes.

We have checked all instances where lake names appear in the text and have made sure the naming is now consistent throughout.

We disagree with Reviewer 1 that the climate analysis is inconclusive. We also feel that the findings are important enough to warrant inclusion in the main text. We clearly show statistically significant association of sediment inferred P yield with climate, associations that vary through the Holocene. That the association with runoff is similar to that observed with rivers is an important observation, and given current interest in temperature control over landscape process it is important that we explain that a direct effect is not seen. We accept, however, that these points were not reflected in the conclusions and have rectified this.